# SEDRAS: Symbolically Evaluated Deep Research And Science

Fredrik Carlsson [* 1]   Daniel Ward [2]   Joseph Ortiz [3]   Fangyu Liu [4]   Joakim Nivre [5]

## Abstract

As the reasoning capabilities of Large Language Models (LLMs) expand, evaluating true inductive generalization on entirely unseen data becomes increasingly challenging. To this end, we introduce a modular in-context learning evaluation framework, that is scalable and extendable across its separate modules. This is based upon the notion of synthetic scenarios with controllable complexity across three independent axes:
**1)** the logic of the underlying data distribution (UDD) **2)** their projection into diverse representations, and **3)** the interaction dynamic determining how the model accesses and explores the data. For these scenarios, the model is tasked to perform in-context scientific discovery and produce an interpretable theory in natural language that explains the observations. In a separate conversation, the model is then tasked to convert this generated theory into executable code, which can be programmatically compared against the underlying data distribution. Using this modular framework we produce an initial suite of 600 diverse scenarios that we use to evaluate and analyze various state-of-the-art LLMs. Although these experiments show that Gemini 3.0 Pro achieves the best overall score, each model performs the best at different tasks. For example: GPT 5.2 is the clear winner on pure symbolic data, Claude Opus 4.5 is the best at working with files, Gemini is the strongest model for the non-dynamic scenarios, and Grok 4.1 is the strongest model when UDD complexity scales. Furthermore, all models struggle with active exploration and are seemingly incapable of identifying informative data points, resulting in less efficient exploration than a random baseline.

[1] Research Institutes of Sweden [2] Emble AB [3] Google DeepMind [4] META (work done at Google DeepMind) [5] Uppsala University. Correspondence to: Fredrik Carlsson <Fredrik.Carlsson@ri.se>.

*Proceedings of the $43^{rd}$ International Conference on Machine Learning*, Seoul, South Korea. PMLR 306, 2026. Copyright 2026 by the author(s).

## 1. Introduction

As the reasoning capabilities of Large Language Models (LLMs) expand, so must the tasks and frameworks used to evaluate them. Static benchmarks are frequently rendered obsolete, either by rapid increases in model capabilities or data contamination (Deng et al., 2024). Recent research has focused heavily on autonomous AI scientists, as well as "deep research", where models analyze vast quantities of data to present coherent findings to humans (Du et al., 2025). However, current deep research benchmarks rely on fixed datasets vulnerable to data leakage, and focus exclusively on analyzing existing data. On the other hand, benchmarks for scientific inference may provide interactive environments (Zheng et al., 2025), but are typically tied to the numerical modality. Alternatively, symbolic math problems are mapped into textual templates (Mirzadeh et al., 2025).

A robust AI scientist must be capable of both synthesizing multi-modal data and engaging in active inquiry by deriving underlying principles through targeted experimentation. Therefore, we introduce a *bottom-up* deep research evaluation framework targeting in-context scientific discovery and the generation of human-interpretable theories of these discoveries. The framework allows for controllable complexity across three independent axes: **1)** the logical structure of the underlying data distributions (UDDs),
**2)** their projection into diverse representations, and
**3)** the dynamics of the interaction protocol, determining how the model accesses and explores the data. Decoupling these axes, means we retain full control of the UDD while the model can still be exposed to the layered complexities of real-world scenarios, with texts, files, tool calls, etc.

Within these environments, the agent is tasked with synthesizing its findings into a comprehensive natural language theory. To evaluate the generated theory, regardless of the data representation, the framework employees a representation-informed translator model that compiles the theory into executable code which is then evaluated against the ground truth UDD. As an initial comparable benchmark, we expect the model being tested to independently act as *all* different LLM components of the evaluation pipeline. For our initial evaluation scenarios, this includes converting the theory into code, and converting natural language queries into variable assignments that can be executed as UDD experiments.

*Table 1.* Comprehensive comparison of our framework against current state-of-the-art benchmarks. Categorical highlighting indicates alignment with robust evaluation requirements (Interactive, Controllable, and Dataset Contamination Resistant).

| BENCHMARK | INTERACTIVE | CONTROLLABLE COMPLEXITY | CONTAM RESISTANT STRATEGY | GENERATION OF NEW TASKS | MODALITY |
|---|---|---|---|---|---|
| AI FEYNMAN | ✗ | ✗ | NONE | NO | NUMERICAL |
| EMPIRICALBENCH | ✗ | ✗ | NONE | NO | NUMERICAL |
| LSR-TRANSFORM | ✗ | ✗ | NONE | NO | NUMERICAL |
| NEWTONBENCH | ✓ | ✓ | PROCEDURAL | LIMITED | NUMERICAL |
| GAIA | ✓ | ✗ | NONE | NO | TEXT / TOOLS |
| WEBARENA | ✓ | ✗ | PERIODIC RESET | NO | WEB / TEXT |
| AGENTBENCH | ✓ | ✗ | NONE | NO | OS / DB / CODE |
| OSWORLD | ✓ | ✗ | SNAPSHOTS | NO | DESKTOP / OS |
| AUTO-BENCH | ✓ | ✓ | DYNAMIC | LIMITED | ADJACENCY MATRIX |
| DEEPRESEARCH | ✓ | ✗ | NONE | NO | WEB / PDF |
| RESEARCHER | ✓ | ✗ | NONE | NO | WEB / TEXT |
| REPORT BENCH | ✓ | ✗ | NONE | NO | TEXT / PDF |
| HYPOBENCH | ✗ | ✓ | PARTIALLY PROCEDURAL | INFINITE | TEXT |
| **OURS** | ✓ | ✓ | **PROCEDURAL** | **INFINITE** | **MULTIMEDIA** |

As an initial starting point we release SEDRAS-2026, consisting of 600 task scenarios that fit within state-of-the-art (SOTA) model context windows. Via evaluation of various SOTA LLMs, we find Gemini-3.0 Pro (DeepMind, 2025) to perform the best, followed by Claude Opus 4.5 (Anthropic, 2025). Analyzing these results further we find different strengths and weaknesses of the models, such as Grok's (xAI Team, 2025) strong performance on complex UDDs, Gemini distinguishing itself on handling textual data, GPT 5.2 (OpenAI, 2025) being the best at raw symbolic data, and Claude's strong ability to work with symbolic solver tools. However, we also discover that all models perform poorly on self-guided scientific discovery, performing worse than simple random exploration. All our results, along with data and code, are available at github.com/FreddeFrallan/SEDRAS.

## 2. Related Work

**The task of symbolic regression** and discovering mathematical expressions from data has emerged as a popular evaluation of LLMs' reasoning capabilities. Benchmarks like AI Feynman (Udrescu & Tegmark, 2020), Empirical-Bench (Cranmer, 2023), and LSR-Transform (Shojaee et al., 2025) present models with numerical tabular data derived from physical laws. However, due to the inherent risks of static datasets, NewtonBench (Zheng et al., 2025) expands upon this by introducing procedural "metaphysical shifts" to create a more resilient benchmark.

**Interactive and agentic** paradigms now serve as a vital cornerstone in modern LLM evaluation. For example, GAIA (Mialon et al., 2024) focuses on complex tool use and reasoning. AgentBench (Liu et al., 2024) provides diverse environments covering areas such as databases and operating

systems. WebArena (Zhou et al., 2023) offers containerized web environments for assessing browsing and interaction. OSWorld (Xie et al., 2024) serves as a platform for testing agents on open-ended desktop and system-level applications. *AutoBench* (Chen et al., 2025) constructs interactive graph-based discovery tasks.

**Deep research** has similarly exploded in popularity, challenging models to analyze vast quantities of, often web data, and synthesize human-readable reports(Wei et al., 2025). Evaluation benchmarks for this include *DeepResearch Bench* (Du et al., 2025) which tests citation accuracy on PhD-level tasks across 22 disciplines, whereas *Researcher Bench* (Xu et al., 2025) evaluates the ability to produce hypotheses for scientific goals. *Report Bench* (Li et al., 2025) prioritizes the synthesis stage, providing rigorous metrics for the factual grounding and structural integrity of the final generated document. Finally, *HypoBench* (Liu et al., 2026), generates rule-grounded datasets for hypothesis discovery, represented via textual templates.

**Our framework** lies at the intersection of these three aforementioned categories (See Section 5.2 for relation to Deep Research). Similar to HypoBench, our bottom-up pipeline generates rich scenarios from fundamental logical components. However, our modularization enables independent scaling of complexity along the framework's different dimensions, whereas previous frameworks simply provide a single dimension at which difficulty can be scaled, if any at all. Our framework can therefore provide fine-grained analysis across independent dimension and their combinations. For example, SEDRAS is the first scientific-oriented framework to **combine** rich data representations and interactive scenarios. Table 1 juxtaposes our framework against previous work, with further discussion in Section 5.

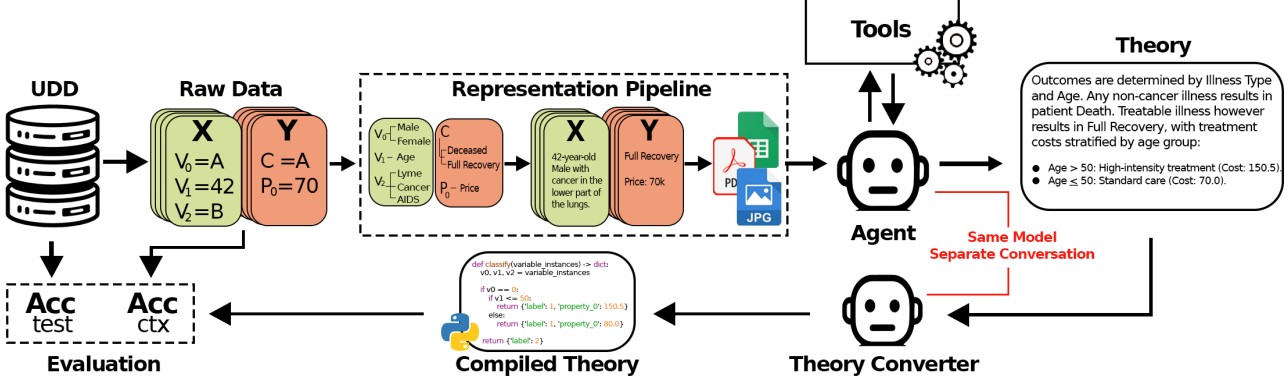

*Figure 1.* Overview of the evaluation framework illustrating the flow from the underlying data distribution through representation transformations, prediction by the agent, theory conversion and final evaluation.

## 3. The Evaluation Framework

Our modular evaluation framework is centered around a controllable *underlying data distribution* (UDD) that serves as the logical ground truth. This abstract data is projected into diverse representations and interactive scenarios, effectively decoupling the core deductive logic from its presentation. This modularity ensures that complexity can be controlled along separate axes. For instance, a task can feature a trivial logical distribution (low UDD complexity) hidden within a highly obfuscated narrative (high representation complexity) and accessed statically (low dynamical complexity).

A model being evaluated is tasked to articulate a theory that explains the underlying data distribution, while simultaneously and independently acting as all LLM components of the scenario. Besides the instruction that the generated theory should human-interpretable, no format constraints are enforced, meaning it may contain any text, pseudo-code, equations, etc. The final step for the model is to independently convert the generated theory into a Python function that matches the UDD format. Finally, this function is programmatically evaluated against the ground truth UDD data.

### 3.1. The Underlying Data Distribution

The UDD serves as the deterministic logic engine of our framework, mapping a set of input variable assignments $X_n$ to a target output $Y_n$. The input space consists of variables typed as either **categorical** (discrete) or **numerical** (continuous spans). The output $Y_n$ is similarly composite, containing a primary class label $C_n$ and optional auxiliary properties, allowing for multi-faceted prediction tasks. (See Appendix B.3 for more information)

The mapping complexity is controlled by two logical primitives: **Variables**, which contribute independent linear effects, and **Rules**, which model non-linear interactions and exceptions. By modulating the rule density and variable

dimensionality, we can precisely scale the logical depth of the benchmark. This structure provides a transparent ground truth for verifying model reasoning, with the formal inference mechanics and voting aggregation detailed in Appendix B.

---

**Examples of Representations**

**Raw**

> **V0:** 2 **V1:** 2 **V2:** 2 **V3:** 0
> **Label:** 0

**Text**

> A **Hardware Malfunction** at the **Local Cell** caused a **Low Impact** network incident, resulting in services becoming **Voice Only** for a brief period.
>
> **Label:** Service Alert

**Text Long**

> Our non-profit organization is excited to announce that our new grant project is officially entering its **Implementation Phase**. This ambitious initiative is specifically designed to achieve significant **Economic Empowerment** for families in the region, offering vital support and training. Although the initial rollout requires careful planning, we currently categorize its overall urgency as **Medium Priority** to ensure a sustainable and robust foundation. We envision this project as a stepping stone towards considerable **Program Expansion**, enabling us to broaden our reach and deepen our impact over the next few years. We believe this effort will transform lives by fostering genuine **Economic Empowerment**.
>
> **Label:** Approved Grant

*Figure 2.* Examples of Raw, Text, and Text Long representations for the same sample. The bold words indicate mapped variables.

*Figure 3.* Illustrations of the three tools available at different dynamics complexities for the agent. The direct experiment tool demonstrates the difference in how it is handled for RAW representations and textual representations.

## 3.2. Representation

The *representation* module determines how the logic from the UDD is perceived by the subject model. In this work, we focus exclusively on textual representations, transforming the raw variable assignments $X_n$ into natural language descriptions. These textual representations are then potentially further rendered as one or multiple files.

Transforming UDD samples into their textual representation is a two-step process. First, a synthesizer model defines a **Semantic Mapping** $\mathcal{M}$. This involves selecting a coherent domain theme and assigning thematic aliases to the UDD variables (e.g., mapping $\text{Var}_0 \rightarrow$ *Blood Pressure*, $\text{Var}_1 \rightarrow$ *Age*). Second, individual samples $X_n$ are converted into text snippets using the synthesizer model and $\mathcal{M}$. Details and prompts for this step are available in Appendix C.1.

Textual samples may further be rendered as files, by using the synthesizer model to dynamically generate a file layout, and populating this layout with samples. In this work we exclusively generate PDF and XLSX files. More information on rendering samples as files is available in Appendix C.5. To steer the complexity of the representations we implement a tiered system. These tiers are illustrated in the lower part of Figure 3, and range from the raw UDD variable assignments to long textual representations rendered in multiple files. Figure 2 contains examples of the non-file representations, and more examples are available in Appendix E.

## 3.3. Dynamics

The *dynamics* module governs the interaction protocol through which the agent acquires information about the environment, and what tools it has available. As an initial starting point for this module, we implement three LLM tools that are visualized in the upper part of Figure 3. Using these tools, we define three settings that scale the dynamic complexity from passive ingestion to active exploration with access to symbolic regression solvers. These are defined as follows:

1. **Static.** The model is provided with a fixed dataset $\mathcal{D} = \{(X_i, Y_i)\}_{i=1}^N$ entirely within its initial context window, and is expected to directly respond with the textual theory.

2. **Active Experimentation.** The model begins with a small set of 10 samples in the context and must actively query the environment via two provided tools: `run_experiment(X)`, which acts as an oracle returning the ground truth output $Y$ for a chosen input, and `evaluate_theory(T)`, which returns the scalar performance metric $Acc_{test}$ for a proposed natural language theory.

3. **Symbolic Solvers.** Access to both the tools from the Active Experimentation setting, along with the access to symbolic solvers from the scikit-learn library (Pedregosa et al., 2011). These solvers take tabular data as input and return either an equation for a numerical output or a decision tree in the form of Python `if-statements` for categorical classification.

Each dynamic experiment also defines an "experimental budget", where the direct experiment costs 1, evaluate theory costs 5, and symbolic regression costs 1. Furthermore, as visualized in upper left part of Figure 3, for scenarios with textual representation, the direct experiment tool utilizes an internal translation LLM. Similar to the theory converter, the model being evaluated also acts as the translation model for this tool. Using $\mathcal{M}$, this tool model translates natural language queries into corresponding UDD representations, executes the experiment, and translates the output back to natural language. This way the main agent is able to perform direct experiments at any representation level. Further details for all tools are available in Appendix D.

## 3.4. The Evaluation Loop

To accommodate the unconstrained nature of LLM reasoning, our framework does not enforce a rigid output schema. Instead, the subject model is prompted to articulate its final hypothesis as a free-form natural language theory $T$, where it is informed that a subsequent model will interpret it as an actionable theory. This theory may therefore encompass prose, mathematical notation, or code snippets. The only constraint is that $T$ must be sufficiently precise to be actionable by the theory converter.

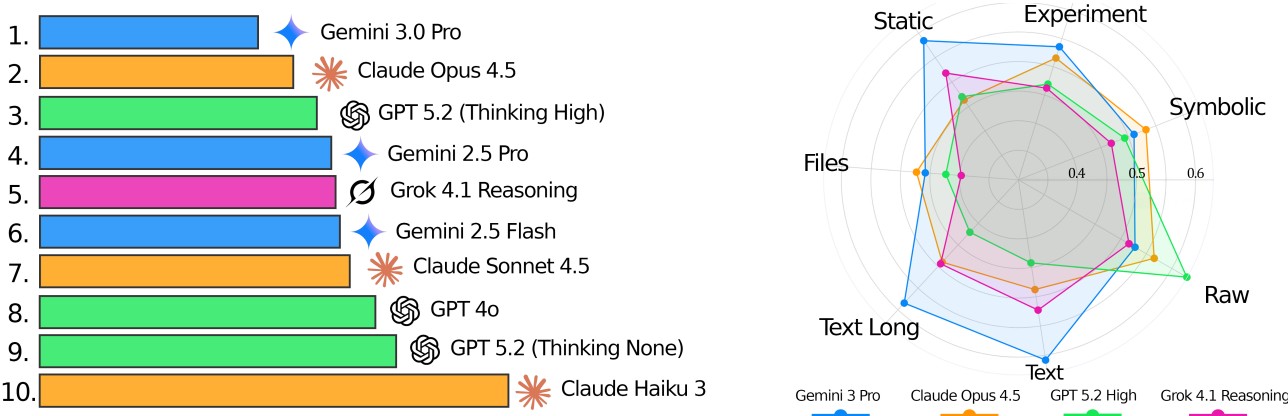

*Figure 4.* **Left:** The relative ranking performance of models on SEDRAS-2026, where lower is better. **Right:** Radar chart displaying the fine-grained analytics of the corresponding results.

To quantify the validity of $T$, we employ a metadata-informed *translator* model which is tasked with converting the abstract theory $T$ into a deterministic Python classifier $f_T(x)$. This translator is provided with the specific Semantic Mapping $\mathcal{M}$ used during generation, allowing it to interpret the theory's domain-specific terminology into the concrete variable space of the UDD.

As elaborated upon in Appendix B.4, $f_T$ is executed against ground-truth data sampled from the UDD, giving us: **In-Context Accuracy** ($Acc_{ctx}$), which represents the accuracy of $f_T$ evaluated on the specific set of samples $X_{seen}$ that the model encountered during its exploration, and **Test Accuracy** ($Acc_{test}$), which measures the accuracy of $f_T$ on a held-out set $X_{test}$ drawn from the same UDD.

As mentioned in Section 3.1, the framework supports scaling the UDD complexity to multi-output classification, where a sample output may contain optional auxiliary classes, along with its primary class label $C_n$. Therefore, accuracy is calculated as the ratio of correct matches to the maximum number of labels present in either the ground-truth or predicted sets, effectively penalizing both omissions and hallucinations. A detailed explanation of this is available in Appendix B.4.

### 3.5. SEDRAS 2026

We release **SEDRAS-2026** as an initial comparable benchmark, consisting of 600 task scenarios evenly spread out across the different scenario settings. Each of these scenarios is created independently, meaning they have their own UDD initialization (see Appendix B.6), and their own textual theme (see Appendix C.2).

To ensure a good spread we combinatorially pair the different complexities of the UDD, representation and dynamics modules. Although the UDD complexity can in practice be scaled indefinitely, we define five complexity levels as specified in Appendix B.5. Throughout these levels there is an increase in the number of underlying variables and rules, and level 4-5 also introduce additional output

properties (See Appendix B.3). Furthermore, we define $N_s \in \{10, 20, \ldots, 100\}$ as the sample budget, representing 10 equidistant steps across this range. For scenarios involving static dynamics, $N_s$ denotes the number of initial samples provided in the context; otherwise, it represents the experimental budget (not counting the 10 initial context samples). Combining all different complexity categories, and the 10 intervals for $N_s$, results in our 600 scenarios.

## 4. Experiments

In the following sections we evaluate a cohort of state-of-the-art LLMs available via their official APIs. From Section 4.1 we distinguish a set of top contender models, which are then the focus for subsequent experiments. Namely, **Gemini 3.0 Pro**, **GPT 5.2 High**, **Claude Opus 4.5**, and **Grok 4.1 Reasoning** (although Gemini 2.5 Pro ranks slightly higher). For the sake of brevity all experiments following Section 4.1 refer to these models simply by their base names.

### 4.1. Overall Evaluation

Due to the evaluation task's exploratory nature, it is difficult to assess how good an absolute $Acc_{test}$ value is, given that particular scenario. Therefore, our main results use relative ranking performance, so to not tempt readers to expect a 100% $Acc_{test}$ across all 600 scenarios. For each scenario, models are assigned a zero-based rank (0 for the top performer, 1 for the runner-up, etc.), with the cumulative sum across all scenarios being the final metric. Consequently, a lower score indicates superior relative performance. However, the absolute values are available in Appendix A

From the results illustrated in the left part Figure 4, we observe that Gemini 3.0 Pro performs the best in terms of relative model capabilities. This is then followed by Claude Opus 4.5 and GPT 5.2 with thinking set to high. In the next tier we have Gemini 2.5 Pro, Grok 4.1 reasoning, and Gemini 2.5 Flash, who all perform fairly similar. Finally, we find the faster/smaller versions of GPT and Claude.

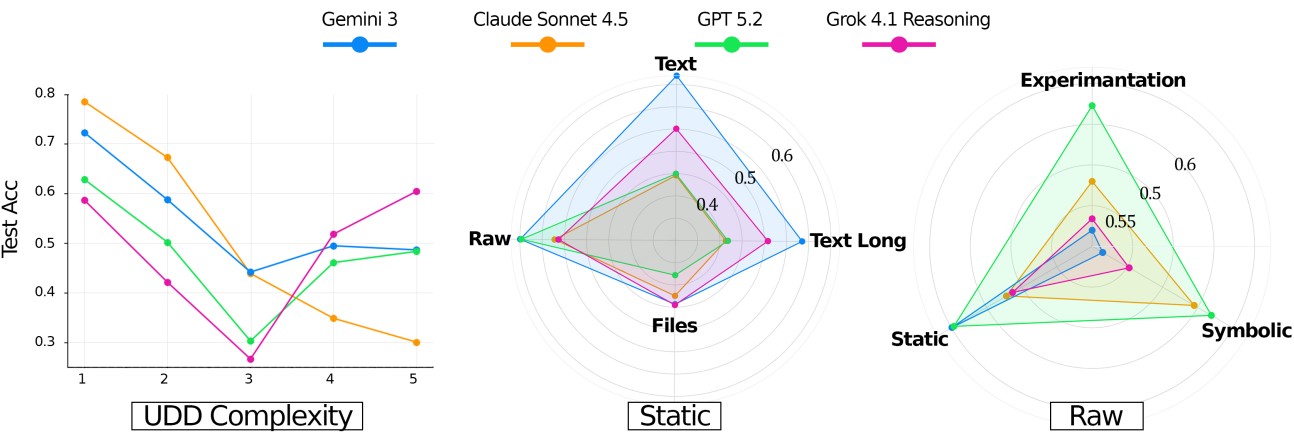

*Figure 5.* **Left:** Graph showing the model performance as UDD complexity scales. **Middle:** Model performance across different representations when the dynamic is set to static. **Right:** Model performance of different dynamic settings for the raw UDD data.

## 4.2. Fine-grained Performance Diagnostics

Using the experiments conducted in Section 4.1 on SEDRAS-2026, we further investigate the models' performance along the different dimensions of the framework. The radar plot in the right part of Figure 4 displays the $Acc_{test}$ for samples belonging to various representation and dynamic settings. These results indicate that Gemini's lead stems mainly from its performance on static samples, and both non-file textual representations. Indeed, Gemini performs noticeably worse on the Raw representations. Claude is the only model that performs worse in the static setting than both interactive settings, but is the best at files with a small margin. Grok sees the largest performance drop when working with files. Interestingly, GPT performs the best on the RAW representations with a clear margin, but it is the worst model for the textual representations.

Figure 5 contains additional plots, one for the UDD complexity (left), representation levels while dynamics is fixed to static (middle), and different dynamic settings while representation is fixed to raw (right). In terms of UDD complexity, Claude performs the best at lower levels and Grok the best at the higher levels, despite Gemini's best overall performance. For the static setting, Gemini mainly distinguishes itself on both textual representations. Finally, from the raw plot we see that GPT is the only model that performs equally well on all three settings.

## 4.3. Scientific Exploration

In this section we isolate the models' ability to perform inductive reasoning through active data acquisition. Using the results from Section 4.1 we specifically investigate the scenarios where the Representation is **Level 1 (Raw)**, and the Dynamics is set to **Active Experimentation**. This ensures that the direct experiment tool is guaranteed to always work, as no model is needed to translate textual queries. We convert each model's experimentation runs into a static counterpart, and build a model-specific **Post-Static** dataset.

This allows us to disentangle the information collection process from any potential side-effects of generating a theory after an interactive conversation with tool usage. This means that the $N_s$ collected samples during each experimentation run is combined with the initial 10 context samples, and then fed all at once to the model in a new separate context. As seen in Figure 6 all models perform significantly better in the static setting, with GPT reaching the highest score of 0.75 at 70 samples. At 100 static samples, all models achieve a score above 0.5, with Gemini above 0.6, and Claude almost 0.7. However, in the post-static setting, all models achieve significantly lower scores, with all results below 0.6 regardless of the number of experiments executed, and often below 0.5.

This clear downgrade in generalization performance, when the models actively choose which inputs to query, reveals an active exploration deficit. Current SOTA models are therefore seemingly incapable of identifying which data points that would be informative, and explore less efficiently than simply randomly selecting data points. Considering that this drop in performance is clearly available for the Post-Static setting, we can infer that this is not due to an inability to understand the data, but rather an inability to strategically collect new data.

## 4.4. Sub-Task Accuracy

Our evaluation framework is designed to analyze a model's general capabilities, and how these change across varying complexities. However, if a mistake is made during the evaluation, this setup does not allow for determining where exactly that error occurred. Therefore, to get a sense of the models' ability to perform the auxiliary tasks, we test the consistency of the theory conversion and the accuracy of the direct experiment with textual representations.

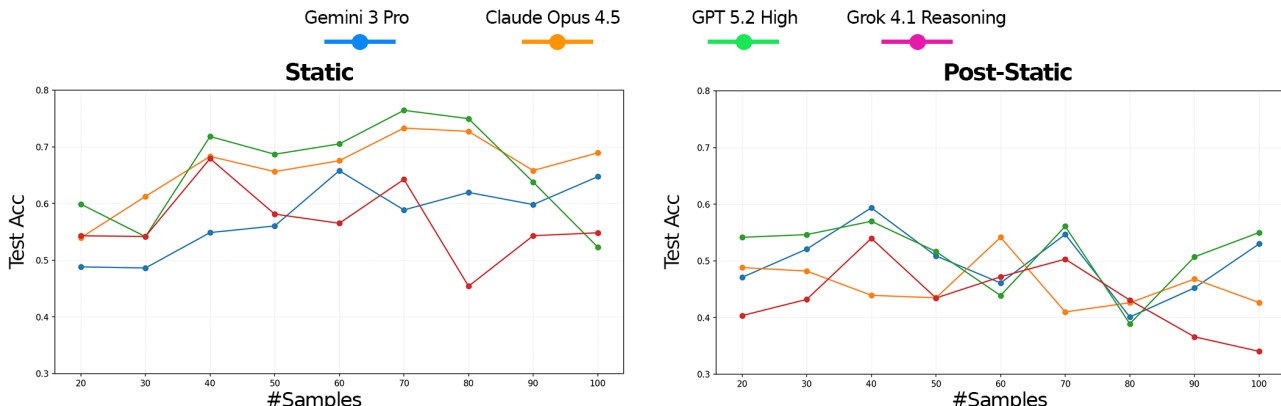

*Figure 6.* **Left:** Test accuracy over the number of in context samples, for the static setting and raw representation. **Right:** Test accuracy when statically evaluated on examples collected from experimentation.

#### 4.4.1. TEXTUAL EXPERIMENTS

For evaluation of the textual query conversion we are able to use the already existing samples from the dataset, where we know the exact variable assignment that corresponds to the textual representation. Therefore, we randomly select 500 textual samples from SEDRAS-2026, and test each model's accuracy on mapping those into the correct original variable assignment. The setup for this conversion is identical to the experiment worker used during evaluation, both in terms of the prompt and its access to the corresponding $\mathcal{M}$. See Appendix D.1 for more details.

Table 2 contains the conversion accuracy for different numbers of variables embedded in the corresponding sample. The general trend is that models perform better with fewer variables, however the rate at which the accuracy drops as the number of variables increases varies greatly. For example, Gemini sees a 5.9% drop from 4 to 7 variables and Claude drops 28.9%.

#### 4.4.2. THEORY CONVERSION

For the theory conversion we do not have access to ground truth pairings of textual theories and python code. Therefore, to provide a sense of how much the models differ in this regard, we instead measure the cross-model consistency when converting theories. For this purpose we randomly select 100 generated theories from the experiments of Section 4.1, and perform theory conversion using each model. We then programmatically compare the resulting Python

classifiers across the different models, and measure the ratio of samples that result in an identical output label.

As seen in Table 3, the compared models produce theories that on average align 91.56% of the time. In particular the two top performing models: Gemini and Claude have over 91% alignment. As can be seen in bottom row of Table 3, the actual difference in downstream accuracy between the models is even smaller since most scenarios are multi-category classification. For example, although Gemini and GPT differ over 11% in their exact alignment, the corresponding difference in accuracy is 7.1% and the difference between Grok and Gemini is only 3.5%.

## 5. Discussion

This work establishes a modular foundation for generating datasets that can evolve alongside AI's increasing capabilities in domains such as scientific discovery and deep research. Through our bottom-up generative pipeline we are able to create data with precise control of the complexity across the framework's three core dimensions independently. This approach provides both the option of fine-grained analysis and a strategy against training data leakage and contamination as a new benchmark instance can be procedurally generated at any point.

*Table 2.* Textual query conversion accuracy (%) per number of variables embedded in the query for the textual experiment.

| Model | 4 | 5 | 6 | 7 | (avg) |
|---|---|---|---|---|---|
| Gemini 3 | 95.8 | 93.3 | 92.6 | 89.9 | 92.8 |
| Grok 4.1 | 85.4 | 88.9 | 82.8 | 79.8 | 83.8 |
| GPT 5.2 High | 92.45 | 88.8 | 83.1 | 73.8 | 84.2 |
| Claude Opus | 98.1 | 81.63 | 80.4 | 69.2 | 82.0 |

*Table 3.* **Upper:** Theory converted classifier alignment matrix. **Lower:** The corresponding evaluation scores for the converted theories.

| Alignment Scores | | | | |
|---|---|---|---|---|
| **Model** | **Gemini** | **Grok** | **GPT** | **Claude** |
| **Gemini** | *100.00%* | 95.47% | 88.94% | 91.31% |
| **Grok** | 95.47% | *100.00%* | 91.86% | 91.84% |
| **GPT** | 88.94% | 91.86% | *100.00%* | 90.13% |
| **Claude** | 91.31% | 91.84% | 90.13% | *100.00%* |
| **Per-Model Downstream Theory Accuracy** | | | | |
| **Eval Acc** | *63.5%* | 60.0% | 56.4% | 57.1% |

The current implementation of each module represents only a subset of the framework's potential. For example, the UDD can be expanded to include continuous differential equations or probabilistic models; the *Dynamics* can support multi-agent collaboration or containerized file systems; and the *Representation* module can be expanded into various modalities such as image, video, and audio. Furthermore, future work could investigate the effects of noise injection within the UDD, Dynamics, or Representation modules.

While the primary intention of this framework is to evaluate AI capabilities, we also note its potential for data creation. Indeed, we speculate that even in its current initial form, the scenarios would be useful as reinforcement learning environments or as a source for static agentic traces. Besides scientific inference and deep research being desirable tasks in themselves, we hypothesize that specifically training a model to excel at our framework's rich scenarios—both in terms of representation and dynamics—would have beneficial carry-over effects to other tasks.

Despite SOTA LLMs' conversational fluency and high performance on static benchmarks, our results indicate a lack of fundamental scientific thinking required for efficient discovery. Indeed, our observation indicates that rather than seeking falsification to prune the hypothesis space, models appear prone to confirmation bias or domain fixation. This weakness implies a clear direction for future work.

### 5.1. Validity of framework

The evaluation framework is designed so that the model being tested is responsible for operating all the different parts of the evaluation loop. Therefore, if one agrees to the validity of the general flow of information, then one may conclude that the evaluation process in itself is valid. Meaning, if any error arises at any point during an active evaluation, then that is an error of the model being evaluated and should rightfully be reflected in the score.

Another potential concern is the room for error in the data creation process of the static data. Indeed, although considerable efforts went into quality checking the different pipeline steps (See Appendix C.3), since LLMs are included, we cannot guarantee flawless data. However, if any such unintended noise exists, then all evaluated models are exposed to the exact same noise. Therefore, any derived results are comparable and still indicative of a model's capabilities.

Furthermore, we argue that incorporating generative AI in the data creation process is an important step for the longevity of the benchmark. Utilizing LLMs, and other forms of generative AI, means that the quality and expressiveness of the benchmark increases alongside the technical progress of this field. This way the evaluation framework naturally scales alongside the very systems it is designed to test.

### 5.2. Relation to Deep Research

*Deep Research* is a term popularized by a setting/mode in the publicly available ChatGPT and Gemini models. In this mode the LLM iteratively searches for information in files and on the internet in order to answer a given user query. This is intended to handle queries requiring more thorough research and information gathering than common queries, and the output format often contains source references to backup its claims. However, to the best of our knowledge, there is now clear definition of what actually constitutes a Deep Research question, environment, or output. Furthermore, the distinction between regular queries and Deep Research becomes increasingly unclear as the general agentic capabilities of LLMs improve.

The initial SEDRAS scenarios does not support iterative web-searches, nor the validation of citations in reports. However, our framework does provide scenarios where the model needs to interact with files, parse data, run experiments etc. Therefore, although Deep Research is defined more broadly than the current implementation of SEDRAS, we argue that our framework helps evaluate skills that are part "Deep Research".

## 6. Conclusion

In this work, we introduced SEDRAS, a modular and procedurally generated evaluation framework designed to test the limits of LLMs in the realm of scientific discovery and deep research. Our experiments with SEDRAS-2026 reveal that while state-of-the-art models like Gemini 3.0 Pro and Claude Opus 4.5 exhibit strong reasoning in static contexts, they fundamentally struggle with active, self-guided exploration. Notably, the observed performance deficit in experimentation suggests that current models lack the strategic inquiry skills required to prioritize informative data points over random sampling. These findings underscore the necessity of moving beyond static benchmarks toward interactive, multi-dimensional environments that can scale in complexity alongside advancing AI capabilities.

The framework's architecture is intentionally modular, allowing the logical, representational, and dynamical modules to be scaled or replaced independently. We encourage the research community to expand upon these capabilities—such as by integrating multi-modal data streams or more complex symbolic solvers—to ensure the benchmark remains a rigorous testbed for future systems. Such extensions will be vital for maintaining a robust, longitudinal understanding of AI progress, provided they are utilized to benchmark an increasingly diverse array of frontier models.

### Impact Statement

This work focuses on evaluation of existing LLMs and does not introduce new methods. However, we heavily rely on publicly available LLM APIs. These infrastructure systems can contribute significantly to global energy consumption.

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

# A. Additional Experiments

*Table 4.* SEDRAS-2026 Model Performance

| Model | Static | Experiment | Symbolic | Raw | Text | Text Long | Files | Total |
|---|---|---|---|---|---|---|---|---|
| Gemini 3.0 Pro | **58.7** | **53.6** | 51.2 | 52.8 | **60.8** | **58.5** | 45.9 | **54.5** |
| Gemini 2.5 Pro | 52.9 | 47.9 | 45.1 | 52.9 | 50.0 | 48.0 | 43.7 | 48.6 |
| Gemini 2.5 Flash | 45.3 | 50.0 | 49.6 | 56.0 | 48.7 | 47.2 | 41.1 | 48.2 |
| Claude Opus 4.5 | 46.6 | 51.7 | **53.0** | 56.6 | 48.8 | 49.9 | **47.3** | 50.4 |
| Claude Sonnet 4.5 | 46.0 | 48.2 | 48.2 | 49.1 | 49.9 | 46.2 | 44.8 | 47.5 |
| Claude Haiku 3 | 26.1 | 32.0 | 31.5 | 41.7 | 37.8 | 37.7 | 0.2 | 29.9 |
| GPT 5.2 (Thinking High) | 46.8 | 46.8 | 49.5 | **62.1** | 44.2 | 42.1 | 42.4 | 47.7 |
| GPT 5.2 (Thinking None) | 41.2 | 42.1 | 44.5 | 45.0 | 41.9 | 44.6 | 39.0 | 42.6 |
| GPT 4o | 43.0 | 45.9 | 46.8 | 52.2 | 44.7 | 43.3 | 40.7 | 45.2 |
| Grok 4.1 Reasoning | 50.0 | 45.6 | 46.8 | 51.3 | 52.3 | 49.1 | 37.3 | 47.5 |

The results in Table 4 demonstrates that different models have aptitudes in different areas. Gemini 3.0 pro performs the best in regards to both static and experimental setups, whilst Opus 4.5 edges it out in the symbolic setting. In regards to handling of different textualization settings GPT 5.2 Thinking High is the clear victor when handling raw data, with Opus 4.5 being the best at handling files. For both text length settings, Gemini 3.0 Pro is the best. Illustrative of the scoring mechanism is Haiku 3 with regards to files: Claude 3 Haiku does not support files, and thus fails at this task.

## A.1. Significance of results

Throughout all experiments, the evaluation of a theory is performed against 1000 randomly selected test data points from the UDD, along with the observed samples seen during inference. The final reported accuracy in any of our experiments is therefore first averaged over the tests for each individual scenario, and then averaged over the relevant scenarios. For example, there are 200 "STATIC" scenarios in SEDRAS-2026.

To verify the significance and robustness of our reported results, we calculate the statistical significance of a model being 1% better, depending on the number of test cases and the number of scenarios. This is done by performing the random selection of the test cases five times for each configuration, and measuring the standard deviation and the maximum absolute difference between any of the five runs. These results are available in Figure 7, and clearly indicate that, even with only 10 scenarios, a theory that performs 1% better than another theory can be considered significant given 1000 test cases.

Finally, we note that for SEDRAS-2026 scenarios with a low UDD complexity, the total number of available input combinations is fewer than 1000. For example, Level 1 (as seen in Appendix B.5) has 4 categorical variables with a maximum of 4 categories each, resulting in $4^4 = 256$ combinations. Therefore, these scenarios actually exhaustively test the theory against all possible input combinations.

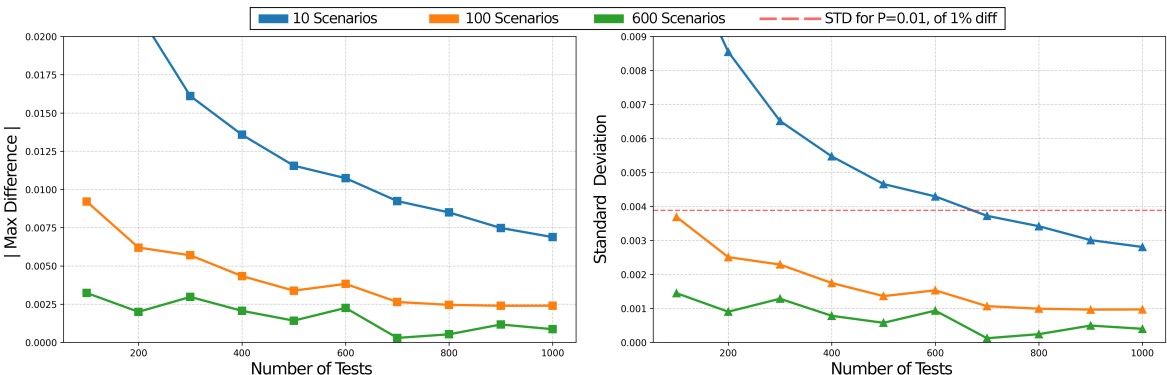

*Figure 7.* Experiments measuring how results differ between 5 evaluation runs, given given the specified number of tests from the UDD. **Left** shows the absolute maximum difference between any of the 5 runs. **Right** shows the standard deviation between the 5 runs, along with the required STD to have a 1% difference in result be statistically significant.

## B. Underlying Data Distributions

The UDD formally defines the variable input space $\mathcal{U}$, where each variable is either a *categorical* (discrete categories) or *numerical* (continuous range). Specifically, $\mathcal{U}$ defines the set of valid discrete categories $\Omega_i = \{s_1, \ldots, s_m\}$ for each categorical variable, and it defines the continuous interval $\Omega_i = [a, b]$ for each numerical variable. We denote $X_n$ as the input data for the $n$-th sample, representing the set of specific variable assignments. Correspondingly, let $Y_n$ denote the target "output scenario," consisting of a primary class label $C_n$ and optional auxiliary class labels referred to as properties (See Section B.3).

Please note that the SEDRAS framework allows for UDDs of formed by many inference mechanisms, rules, and non-linearity. Below, we describe the formation of the SEDRAS-2026 scenarios UDD.

### B.1. Inference Mechanism.

The derivation of the class label $C_n$ is a weighted voting process. Assuming $K$ possible output classes, each potential variable assignment $v$ is pre-initialized with a parameter pair for every label $k \in \{1, \ldots, K\}$: a weight $w_{v,k} \in [0, 1]$ and a score $s_{v,k} \in [0, 1]$. Here, $w_{v,k}$ quantifies the confidence of assignment $v$ regarding label $k$, while $s_{v,k}$ represents the magnitude of support.

To determine $C_n$, the system aggregates the contributions of all active assignments in $X_n$, selecting the label that maximizes the normalized score as defined in Equation 1:

$$C_n = \underset{k \in \{1, \ldots, K\}}{\operatorname{argmax}} \left( \frac{\sum_{v \in X_n^*} w_{v,k} \cdot s_{v,k}}{\sum_{v \in X_n^*} w_{v,k}} \right) \tag{1}$$

### B.2. Rules and Non-Linearity.

To capture feature interactions, the UDD incorporates **Rules**. A Rule is defined as a conditional entity that activates only upon a specific conjunction of variable assignments (e.g., $\text{Var}_A = val_a \wedge \text{Var}_B = val_b$).

Crucially, when a rule is triggered, it **supersedes** the constituent variables. The individual variable assignments involved in the rule are removed from the active set $X_n$ and replaced by the rule's specific weight and score in the effective set $X_n^*$ used in Equation 1. This mechanism enables the modeling of strict overrides, where a specific combination of features negates the predictive signals provided by the features in isolation.

### B.3. Output Properties

Output properties can be thought of as additional output labels that are conditioned on the primary output class. Formally, we define a set of properties $\mathcal{P} = \{P_1, \ldots, P_J\}$. Each property $P_j$ is associated with a fixed set of valid property labels $\Omega_{P_j} = \{p_1, \ldots, p_l\}$ and a subset of primary output classes $\mathcal{C}_{P_j} \subseteq \{1, \ldots, K\}$ for which the property is active. A specific property $P_j$ is evaluated for a sample $X_n$ if and only if the derived primary class label satisfies $C_n \in \mathcal{C}_{P_j}$.

The general inference mechanism for determining the specific property label is identical to the voting process used for $C_n$. For each property $P_j$, every variable assignment $v$ is pre-initialized with a parameter pair for every possible property label $p \in \Omega_{P_j}$: a weight $w_{v,p} \in [0, 1]$ and a score $s_{v,p} \in [0, 1]$. The final property label $P_{j,n}$ is independently determined by the logic of Equation 1, substituting the property label $p$ for the class $k$:

### B.4. Measuring Accuracy

To evaluate the performance of the system, we define a custom accuracy metric that accounts for both the primary class $C_n$ and the set of active properties $\mathcal{P}_n$. Because the number of active properties can vary between the ground truth and the prediction, the accuracy is calculated by normalizing the number of correct matches by the maximum number of potential labels present in either set. As elaborated upon in Appendix A.1, we do this for 1000 data points for each sample.

Let $L_n = \{C_n\} \cup \{P_{j,n} \mid C_n \in \mathcal{C}_{P_j}\}$ be the set of ground truth labels, and let $\hat{L}_n$ be the set of predicted labels. The accuracy for a single sample is defined as:

$$\text{Acc}_n = \frac{|L_n \cap \hat{L}_n|}{\max(|L_n|, |\hat{L}_n|)} \tag{2}$$

This formulation ensures that both missing labels and extraneous "hallucinated" labels are penalized equally. For example, if the ground truth contains a class and one property ($|L_n| = 2$) but the model only predicts the correct class ($\hat{L}_n = 1$), the accuracy is $1/2 = 50\%$. Conversely, if the ground truth contains only a class ($|L_n| = 1$) but the model predicts a class and an additional property ($|\hat{L}_n| = 2$), the accuracy is likewise $1/2 = 50\%$. If the ground truth contains three labels and the prediction matches two of them, the accuracy is $2/3 \approx 66\%$.

## B.5. Complexity Categories

Unlike the the naturally discrete complexity categories of our representation and dynamics implementation, the UDD can in practice be scaled indefinitely. I.e by increasing the number of variables, output categories, output properties, and rules. For the creation of SEDRAS-2026 we therefore define 5 complexity categories as described in Table 5.

*Table 5.* Complexity Categories for SEDRAS-2026

| Metric | Level 1 | Level 2 | Level 3 | Level 4 | Level 5 |
|---|---|---|---|---|---|
| # Numerical Variables | 0 | 1 | 1 | 1 | 2 |
| # Categorical Variables | 4 | 4 | 5 | 5 | 5 |
| # Categories per Variable | [3, 4] | [3, 4] | [3, 5] | [3, 5] | [3, 5] |
| # Rules | 6 | 6 | 8 | 8 | 10 |
| # Output Categories | 2 | 3 | 4 | 4 | 4 |
| # Output Properties | 0 | 0 | 0 | 1 | 2 |

## B.6. Initialization Search

At start, the UDD parameters, such as the rules and weights, are randomly initialized. However, this often leads to an imbalanced class distribution and the majority of samples having the same output label. Therefore, we incorporate a rudimental evolutionary search (Goldberg, 1989) that results in a more balanced dataset. Additionally, we strive to create a dataset where the underlying rules results in conclusive categorizations. Combining these two objectives we get the objective function we optimize towards, as defined in Equation 3.

$$Score = \Delta_{dec} + \delta_{gap} + (w_{prop} \cdot \bar{\rho}_{prop})$$

where:

$$\Delta_{dec} = \max(0, d_{target} - d_{min})$$

$$\delta_{gap} = \max_i \left| f_i - \frac{1}{N} \right| \tag{3}$$

$$\bar{\rho}_{prop} = \frac{1}{|K|} \sum_{k \in K} \delta_{gap,k}$$

$\Delta_{dec}$ calculates the decisiveness shortfall, penalizing instances where the margin between the two highest-scoring labels falls below a target threshold $d_{target} = 0.75$. The term $\delta_{gap}$ represents the global class imbalance, measuring the deviation of any class frequency $f_i$ from a perfectly uniform distribution, across the $N$ available dataset labels. Although the UDD parameters for additional properties are completely separate, we set $w_{prop} = 0.5$ so to prioritize the search towards the overall labels. The search utilizes a population of 20, a mutation rate of 0.2, and the search is executed for 20 generations. We leave it to future research to explore the role of imbalanced datasets in this analysis; this provides another axis by which the complexity of the task can be scaled.

# C. Representations and Transformations

Given the complex nature of the implementation of the project, this appendix contains only a summarized overview of the implementation details. For a full insight we refer to the available code at: **–ANONYMIZED LINK–**.

## C.1. Semantic mapping instances

*Table 6.* The 100 semantic mapping instance hints used for evaluation.

| | | | |
|---|---|---|---|
| 1. Crime investigation | 26. Industrial audit | 51. Nutritionist plan | 76. Symphony analysis |
| 2. Medical diagnosis | 27. Botanical catalog | 52. IP infringement | 77. Resource audit |
| 3. Restaurant review | 28. Grant summary | 53. E-sports bracket | 78. Course syllabus |
| 4. Space mission log | 29. Stage directions | 54. Telecom outage | 79. Cleanroom log |
| 5. Experiment notes | 30. Insurance damage | 55. Museum checklist | 80. Marathon splits |
| 6. Fantasy profile | 31. Crypto whitepaper | 56. Harvest yield | 81. Court transcript |
| 7. Interview transcript | 32. Gym workout log | 57. Phonetic data | 82. Blueprint spec |
| 8. Weather bulletin | 33. Podcast notes | 58. Bug submission | 83. Recipe instructions |
| 9. Event chronicle | 34. Philanthropic report | 59. Traffic report | 84. Cyber incident |
| 10. Sports commentary | 35. Social media brief | 60. Mineral grade | 85. Earnings summary |
| 11. Election analysis | 36. Return reasoning | 61. Rule clarification | 86. Shipping manifest |
| 12. Attendance log | 37. Debate record | 62. Assembly step | 87. Production notes |
| 13. Relationship diary | 38. Veterinary record | 63. Cupping notes | 88. Zoning report |
| 14. Military debrief | 39. Concierge log | 64. Enviro. study | 89. ER triage notes |
| 15. Quest description | 40. Star chart | 65. Mood board | 90. Stock analysis |
| 16. AI diagnostic | 41. Transport survey | 66. Ledger entry | 91. Aerospace stress |
| 17. Personality assessment | 42. Biotech lab log | 67. A/B test results | 92. Poetry breakdown |
| 18. Plot synopsis | 43. Brand guidelines | 68. Flight pre-check | 93. API documentation |
| 19. Excavation record | 44. Disaster plan | 69. Migration study | 94. Progress report |
| 20. Travel journal | 45. Festival metadata | 70. Clinical protocol | 95. Thesis abstract |
| 21. Real estate listing | 46. Library conservation | 71. Waste efficiency | 96. Patent description |
| 22. Automotive repair | 47. Yoga itinerary | 72. Census breakdown | 97. Disruption alert |
| 23. Runway description | 48. Renewable energy | 73. Seating logic | 98. Yoga retreat goals |
| 24. Wildlife notes | 49. Venture pitch | 74. Energy rating | 99. HR review |
| 25. Support ticket log | 50. Deep-sea sonar | 75. Investigator log | 100. Inventory report |

The transformation process of UDD samples follows a multi-step process, where each step in itself is simple and robust. **1)** An overarching theme is selected for the dataset. **2)** Each variable and its potential classes are mapped to textual descriptions. **3)** Using the textual corresponding descriptions, each sample is transformed into natural text snippet using an LLM. **4)** The textual samples are potentially rendered as files.

As specified below, certain steps of this process utilizes a dataset creation LLM, which we refer to as the *synthesizer model*. Unless specified otherwise, the synthesizer model was chosen to be "Gemini 2.5 Flash" for all our created datasets. However, each task performed by the synthesizer model is in general trivial, and could therefore be done by any LLM. The only exception we encountered being generating file-template code as discussed in C.5, where we instead used Gemini 3.0.

Although the incorporation of LLMs introduce a level of uncertainty in the generation process, we implement programmatic checks at various places in the creation pipeline. Namely, the textual transformations are validated to contain the correct target keywords, and the file representations are validated to contain all target samples. More on this in Section C.3 and Section C.5.

## C.2. Textual Theme and Semantic Mapping

The first step in any transformation is to select an overarching theme. To ensure a good spread among the different datasets, we keep a pre-generated list of 100 unique themes and uniformly sample from this list. The full list of themes is available in Table 6.

After a theme has been selected, the synthesizer model is tasked with generating the semantic mapping $\mathcal{M}$. This way each variable, and its potential subcategories, are mapped to text descriptions that fit the target theme. Concretely, $\mathcal{M}$ is generated by providing the model with the UDD variable input space $\mathcal{U}$, the target theme, and the instruction prompt defined in C.4.

## C.3. Textual Transformations

Given the mapping $\mathcal{M}$, each sample is individually transformed into a textual representation using the synthesizer model. The prompt used for this depends on the target representation, which for our paper means either "Text" or "Text Long". Both of these prompt are defined in F.2.

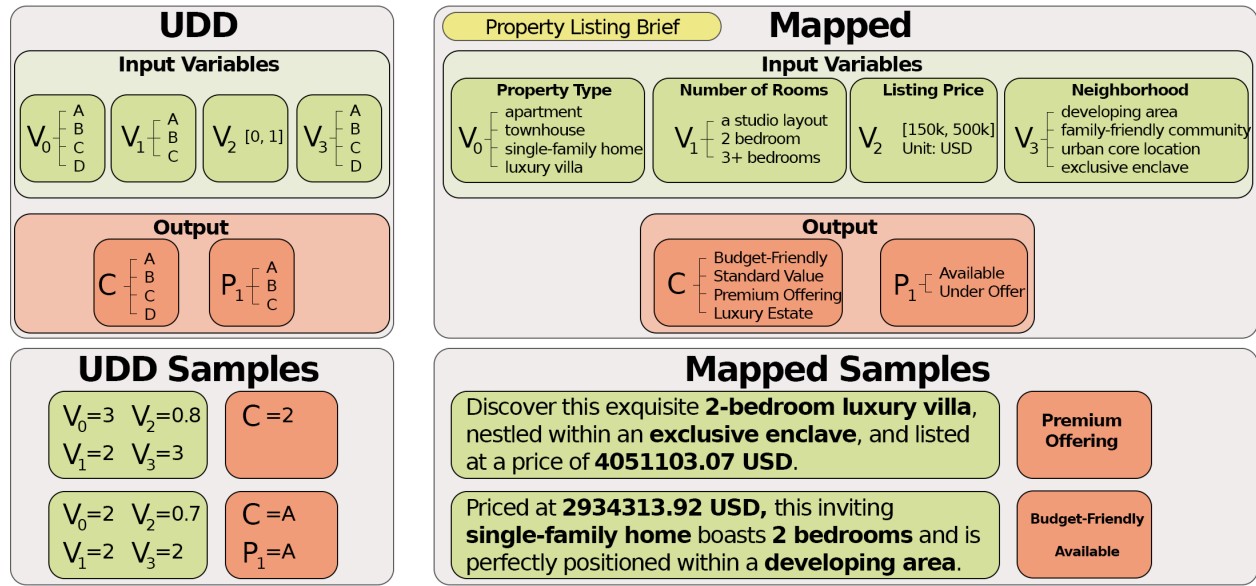

*Figure 8.* Visualization of how the semantic mapping is used for the creation of textual samples.

To provide at least rudimentary guarantees during data creation, we chose to rely on fixed keywords for each variable mapping. This means that the exact syntax and string of a variable is expected to occur after any textual transformation. Therefore, we are able to programmatically validate a generated text snippet by checking that it contains all expected sub-strings, and if they do not we simply rerun that transformation step.

Notably, the task of generating a short text snippet that contains a set of keywords is far below the capabilities of today's LLMs. Therefore, we encourage future work to investigate more complex textual transformations that go beyond fixed keywords.

### C.4. Textual Theme and Semantic Mapping Prompt

### C.5. File Representations

Once samples have been converted to a textual representation, they may further be rendered as PDF and XLSX files. This is achieved by having the synthesizer model generate a file layout that includes the already transformed samples. While it is possible to render any of the representation as files, we stick with the "Text Long" representation to keep a clear progression in complexity. The structure of this layout is encouraged to be *creative* and divide the samples into various sections, along with adding additional irrelevant textual content.

Instead of having the synthesizer model generate the full file content on its own, it instead generates a general file layout in python code. During this step the synthesizer is provided with an existing tool function that takes a sample ID and returns the textual content of that sample. This means that the file can be populated with the exact textual representation of samples, without any further alterations or work by the synthesizer. Examples of generated layouts can be found in C.6.

By tracking what samples that has been inserted into the file, we are able to programmatically verify that all target samples has been included. Hence, given that the validation steps of the textual transformations passed, we can conclude that the file contains all target samples with a valid textual representation.

When rendering a set of textualzied samples, we first randomly select the target number of files in the range $[1, 5]$. Each of these files is then randomly assigned a subset of the available samples that may be overlapping, but so that each sample occurs at least once. The synthesizer generates a custom layout for all files independently, resulting in the final data representation.

### C.6. File Representations Examples

Below are examples of files that the models worked with. They are taken from two different representations, the PDF is of environmental impact assesments, whilst the XLXS is of renewable energy produced on a farm.

## D. Dynamics and Tool Usage

### D.1. Direct Experiment Tool

**Example of Raw Experiment**

**LLM Agent Output**

. . .

There seems to be a clear pattern emerging from the data. However as we still have budget for 3 more experiments, let me run some final experiments.

**Experiment Tool Call**

**Assignment:** $\{V0 : 0, V1 : 2, V2 : 2, V3 : 0.64\} \rightarrow \{label : \mathbf{2}, property1 : \mathbf{3}\}$

**Experiment Tool Call**

**Assignment:** $\{V0 : 4, V1 : 1, V2 : 1, V3 : 0.4\} \rightarrow \{label : \mathbf{1}\}$

Given these results, and that we have used up our experimental budget, here is my final theory:

. . .

**Example of Textual Experiment**

**LLM Agent Output**

I suspect there is strong correlation between the samples which contain "Introverted", "Intuitive", and the profiles being labeled as "Developing Profile". Let me run an additional example to verify my hypothesis.

**Experiment Tool Call**

"*An introverted individual with an expressive attitude and a balanced CV has entered the recruitment pool.*"

**Textual Experiment Input**

```
You are a conversion assistant.  Convert the user's natural language
experiment request into a JSON object mapping each variable name to an integer
category.
```

**output labels:** 0:"*Developing Profile*", 1:"*Established Profile*", 2:"*Distinct Profile*"
**V0:** 0:"Introverted", 1:"Ambivert", 2:"Extroverted", 3:"Fluid"
**V1:** 0:"Analytical", 1:"Intuitive", 2:"Balanced",
**V2:** 0:"Reserved", 1:"Responsive", 2:"Expressive", 3:"Volatile"

**Query:** "*An introverted individual with an expressive attitude and a balanced CV has entered the recruitment pool.*"

**Experiment Execution**

**Assignment:** $\{V0 : 0, V1 : 2, V2 : 2\} \rightarrow \{label : \mathbf{2}\}$

**Textual Experiment Response**

$\{label : \textbf{Distinct Profile}\}$

### D.2. Evaluate Theory Tool

Following is an example of the evaluate theory tool:

---

**Example of Theory Evaluation**

**LLM Agent Output**

. . .

After analysis, the key pattern is: $V1 > V0 + V3 - V2$. This simplifies to: $V1 + V2 > V0 + V3$

Checking all cases confirms this works for all samples seen so far.

Let's evaluate our theory.

> **Theory Evaluation Tool Call**
>
> *The outcome is Label 1 if the sum of V1 and V2 exceeds the sum of V0 and V3; otherwise, the outcome is Label 0.*

**Theory Evaluation Input**

Convert the following THEORY description into a deterministic Python classifier.

Emit ONLY one function with EXACTLY this signature:

**def classify(x: list[float])**

- Return a dictionary with an integer entry label under the key **"label"**.
- x[i] is the value for the i-th variable in this fixed order:
- Categorical variables use integer category IDs.
- Numerical variables use floating-point values (e.g., spans normalized to [0, 1]).
- Return exactly one of these integer labels: $\{0, 1\}$
- No prints, no imports, no randomness.
- Implement the logic faithfully; do not add heuristics not stated in the theory.
- Make sure you interpret variables in the textual theory, and convert them to the correct variables and order.

Now output ONLY valid Python code for **def classify(x: list[float])** (no extra text).

> **Theory Conversion Output**
>
> ```python
> def classify(x:  list[float]):
>     V_0, V_1, V_2, V_3 = x[0], x[1], x[2], x[3]
>     if V_1 + V_2 > V_0 + V_3:
>         return {"label":  1}
>     else:
>         return {"label":  0}
> ```

> **UDD Python Evaluation Output**
>
> $\{accuracy : \mathbf{0.75}\}$

**Theory Evaluation Response**

$\{accuracy : \mathbf{0.75}\}$

## D.3. Symbolic Solver Tool

Following is an example of the symbolic solver tool:

---

**Example of Symbolic Solver**

**LLM Agent Output**

**Symbolic Solver Tool Call**

samples = [
   $\{"0" : 1, "1" : 2, "2" : 0, "3" : 1, "label" : "2"\}$,
   $\{"0" : 0, "1" : 0, "2" : 0, "3" : 2, "label" : "0"\}$,
   $\{"0" : 2, "1" : 0, "2" : 1, "3" : 0, "label" : "1"\}$,
   ...
]

**Symbolic Solver Process (No LLM involved)**

**Scikit Learn Decision Tree Solver**

```
|--- V₁ ≤ 0.5
|   |--- V₂ ≤ 0.5
|   |   |--- Class:  0
|   |--- V₂ > 0.5
|   |   |--- Class:  1
|--- V₁ > 0.5
|   |--- V₁ ≤ 1.5
|   |   |--- V₀ ≤ 0.5
|   |   |   |--- Class:  1
|   |   |--- V₀ > 0.5
|   |   |   |--- Class:  0
|   |--- V₁ > 1.5
|   |   |--- Class:  2
```

**Tree To Python**

```
...
```

**Symbolic Solver Response**

```
def predict(x:  list[float]):
    V₀, V₁, V₂, V₃ = x[0], x[1], x[2], x[3]
    if V₁ ≤ 0.5:
        if V₂ ≤ 0.5:
            return "label":  0
        else:
            return "label":  1
    else:
        if V₁ ≤ 1.5:
            if V₀ ≤ 0.5:
                return "label":  1
            else:
                return "label":  0
        else:
            return "label":  2
```

---

# E. Examples of created data samples

## E.1. Raw

**Examples of Raw Samples**

assignment: V0: 2, V1: 2, V2: 2, V3: 0
label: 0

assignment: V0: 1, V1: 3, V2: 3, V3: 2, V4: 2
label: 2

assignment: V0: 0, V1: 4, V2: 2, V3: 0, V4: 3, V5: 2
label: 0

assignment: V0: 1, V1: 0, V2: 3, V3: 3, V4: 2, V5: 0
label: 1

assignment: V0: 3, V1: 2, V2: 0, V3: 0, V4: 0, V5: 2, V6: 0 label: 3

## E.2. Text

**Examples of Themed Text Samples**

**Instance type:** Network Outage Incident Report
**Label:** Service Alert
A **Hardware Malfunction** at the **Local Cell** caused a **Low Impact** network incident, resulting in services becoming **Voice Only** for a brief period.

**Instance type:** Property Damage Claim Summary
**Label:** Unresolved
Following a **Fire Incident** that caused **Extensive Damage** to the **Exterior Grounds**, the claim, with the **Estimated Financial Cost for Repair or Replacement: 146094.43 USD**, has been **Approved for Settlement**.

**Instance type:** Employee Performance Review
**Label:** Satisfactory
An employee's overall performance **Exceeds Expectations**, especially through their **Innovative Approach** and dedication to **Mentoring Others**, though some areas **Require Development**. This individual also displayed **Minimal Engagement** in certain team activities despite having **Total Training Hours Completed: 24.09 hours**.

**Instance type:** Renewable Energy Output Log
**Label:** Warning Issued
Despite a **Stable Grid Connection**, renewable energy output was **Very Low (0–25 MWh)** (**2 MWh**) due to **Overcast Skies** and **High Temperature Stress**, leading to **High Resource Intensity** and the system being in **Emergency Shutdown**.

## E.3. Text Long

## Examples of Detailed Text Samples

**Instance type:** Non-Profit Grant Project Summary
**Label:** Approved Grant
Our non-profit organization is excited to announce that our new grant project is officially entering its **Implementation Phase**. This ambitious initiative is specifically designed to achieve significant **Economic Empowerment** for families in the region, offering vital support and training. Although the initial rollout requires careful planning, we currently categorize its overall urgency as **Medium Priority** to ensure a sustainable and robust foundation. We envision this project as a stepping stone towards considerable **Program Expansion**, enabling us to broaden our reach and deepen our impact over the next few years. We believe this effort will transform lives by fostering genuine **Economic Empowerment**.

**Instance type:** Decentralized Protocol Status Report
**Label:** Moderate
The decentralized protocol is currently making significant strides in **Phase 2: Alpha Development (2.38 Phase)**, with our engineering teams working tirelessly. We are very pleased with the initial performance of our core consensus mechanism, a robust **Proof-of-Stake (PoS)** system, which shows great promise. During recent internal audits, **Minor Vulnerabilities Identified** were quickly logged for immediate resolution by our dedicated security experts. For now, the system remains an **Isolated Protocol**, allowing us to thoroughly test functionalities without external dependencies or distractions. This controlled environment is crucial for perfecting the economic model, which is built around a **Fixed Supply (Deflationary)** token, ensuring long-term stability and value. We anticipate moving swiftly through the remaining alpha stages.

**Instance type:** API Endpoint Status Report
**Label:** Offline
Our primary API endpoint remains **Stable**, ensuring reliable service for most operations. However, we've noted that certain transaction types are consistently **Slow (200–1000ms)**, impacting user experience. Despite this, our overall error rate remains commendably **Low (0–1%)**, indicating robust internal processing. Regarding resource utilization, while our target for background process RPM is **Very Low (0–50 RPM)**, current telemetry shows activity at **285.36 RPM**, necessitating further investigation. All external access continues to be secured by industry-standard **OAuth 2.0**, with data exchange strictly maintained as **JSON Only**.

**Instance type:** Gym Workout Session Log
**Label:** Planned
Today's gym session had a particular focus on the **Arms**, aiming for endurance rather than heavy lifting. I kept the intensity **Light**, choosing to focus on form and control through each repetition. My energy levels felt **Very Low** during the initial warm-up, making it a bit of a struggle to get started. Despite that, the overall feeling of the workout was **Good**, and I pushed through until the end. I found myself feeling quite **Dehydrated** afterwards, reminding me to increase my water intake. The **Total Workout Duration: 47.4 Minutes** passed quickly, making it a solid, if not super intense, training day.

**Instance type:** Botanical Specimen Record
**Label:** Tropical Tree
This remarkable **Climbing Vine** unfurls its delicate foliage each **Spring**, painting the garden with a renewed sense of life. Its leaves, distinctively **Ovate** in shape, provide a lush backdrop for the stunning blossoms that emerge, often a vibrant shade of **Blue**. Gardeners should be aware of its impressive potential, as it can reach an **Average Mature Height: 21.59 Meters**. Thriving best in **Zone 3–5**, this resilient species endures colder climates with grace. For optimal cultivation, an **Optimal Soil pH: 5.49 pH** is crucial.

## ECOAUDIT: ENVIRONMENTAL IMPACT ASSESSMENT ANALYSIS

*A comprehensive review of resource extraction proposals classified by ecological risk factors.*

**Executive Summary**

Distinguishing acceptable from unacceptable environmental impacts in resource extraction projects centers on several key criteria. Acceptability is typically determined by the impact's scope (localized versus widespread), duration (temporary versus long-term or irreversible), and the demonstrated effectiveness of proposed mitigation measures, ensuring compliance with regulatory standards and industry best practices. Unacceptable impacts, conversely, are those that lead to significant, irreversible degradation of critical ecosystems, threaten biodiversity, create unmanageable cumulative effects, or cannot be sufficiently avoided or mitigated to protect environmental integrity and community well-being.

| Acceptable Impact | Unacceptable Impact |
|---|---|
| **0 (Acceptable Impact)** The recent Environmental Impact Assessment meticulously reviewed a proposed Resource Extraction Operation, specifically focusing on its environmental footprint. Particular scrutiny was given to potential disturbances to adjacent Freshwater Systems, a major point of contention during the preliminary stages. The assessment ultimately identified a significant and Critical Environmental Risk if proper safeguards are not meticulously implemented throughout the project's duration. Fortunately, a comprehensive set of proposed mitigation strategies, including advanced treatment protocols, were deemed Likely Effective in minimizing the negative impacts of the Resource Extraction Operation. Protecting the integrity of the Freshwater Systems remains paramount, and ongoing vigilance will be essential. Should these measures be consistently applied, they are considered Likely Effective in maintaining ecological balance, even amidst a large-scale Resource Extraction Operation. | **1 (Unacceptable Impact)** The recent Environmental Impact Assessment for the **Residential Expansion Project** revealed significant concerns. This ambitious development, poised to transform vast swathes of land, has been flagged due to its potential impact on the delicate ecosystems within **Marine & Coastal Zones**, which are vital for local biodiversity and tourism. The assessment clearly indicates a **High Environmental Risk** associated with various stages of construction and long-term habitation. This risk necessitates careful consideration of mitigation strategies, for which preliminary proposals are deemed **Potentially Effective** if implemented rigorously. Project stakeholders now face the crucial task of ensuring these measures are not just theoretical but become practical solutions to safeguard our natural heritage, especially within the vulnerable **Marine & Coastal Zones**. Ultimately, while the **Residential Expansion Project** presents undeniable development opportunities, addressing its **High Environmental Risk** with **Potentially Effective** solutions is paramount for sustainable growth. |

*Figure 9.* Example of a generated PDF layout.

| Sample 4 \| 1 (Operational Event Alert) | Sample 7 \| 1 (Operational Event Alert) |
|---|---|
| Today's log for the renewable energy farm shows promising data regarding **Wind Power**. The morning began with **Overcast/Strong Wind** conditions, which initially hinted at an excellent generation period. Indeed, for several hours, the farm achieved **Optimal Output** from its turbines. However, an unexpected sensor malfunction triggered an **Emergency Shutdown** on the north array. This brief | The Renewable Energy Farm's daily log started with the usual checks, noting the prevailing **Partly Cloudy/Moderate Wind** conditions that kept the turbines active. Early in the afternoon, however, a critical anomaly was detected within the **Hydroelectric** generation unit. This necessitated an immediate **Emergency Shutdown** of that entire section to prevent further issues. While the unexpected halt certainly |
| **Sample 8 \| 1 (Operational Event Alert)** Today's log for the renewable energy farm notes a wonderfully **Clear/Calm** sky, which usually bodes well for our solar array. However, we've encountered some unexpected issues with our **Hydroelectric** component, which has led to some complications across the board. The main challenge today is a reported **Partial System Downtime** affecting a significant portion of our grid | **Sample 9 \| 1 (Operational Event Alert)** Today's log indicates robust **Wind Power** generation early on, thanks to the **Overcast/Strong Wind** conditions. However, a routine check revealed the need for **Minor Component Maintenance** on one of the larger turbines. This unexpected work resulted in a period of **Low Output** from that specific section of the farm. Despite the generally favorable weather for **Wind Power**, the necessary repairs |
| **Sample 11 \| 1 (Operational Event Alert)** The daily log for our renewable energy farm presented a mixed picture today. While the solar arrays struggled under the persistent **Overcast/Strong Wind**, our **Hydroelectric** generators performed admirably, taking much of the strain. Unfortunately, the north section of the wind farm experienced **Partial System Downtime** for several hours due to an unexpected sensor malfunction. This | **Sample 12 \| 1 (Operational Event Alert)** The renewable energy farm reported an intriguing day for power generation. Despite an Overcast/Strong Wind advisory that kept the turbines exceptionally busy, we noted a Partial System Downtime in one of our secondary solar arrays this morning. Nevertheless, the reliable Hydroelectric segment worked tirelessly to maintain balance, ensuring an Optimal Output for the overall grid for most of the daylight hours. The |
| **Sample 13 \| 1 (Operational Event Alert)** The morning started with a beautiful sunrise over the turbines, promising a day of optimal production. Weather conditions were reported as Clear/Calm, which usually bodes well for our solar array and wind turbines. However, we encountered a snag with our Hydroelectric generation unit, which experienced Partial System Downtime for several hours. Despite this setback, the overall farm still | **Sample 17 \| 1 (Operational Event Alert)** Today began beautifully with a Clear/Calm sky, promising a productive day for our solar arrays and wind turbines. However, an unforeseen issue quickly developed at the Hydroelectric plant, requiring immediate attention. By mid-morning, the situation escalated, leading to an Emergency Shutdown of the entire hydroelectric system. Despite the excellent weather conditions for other sources, |
| **Sample 18 \| 1 (Operational Event Alert)** The day at the renewable energy farm began with promising conditions across all sectors. Our **Hydroelectric** division reported steady performance, contributing significantly to the grid. Despite the **Partly Cloudy/Moderate Wind** conditions throughout the morning, our wind turbines managed to achieve near-peak efficiency. In fact, most of our solar arrays reached **Optimal Output** for several hours, | **Sample 19 \| 1 (Operational Event Alert)** Today began with **Partly Cloudy/Moderate Wind** conditions across the solar and wind arrays, setting an interesting tone for energy generation. Unfortunately, our **Hydroelectric** plant experienced **Partial System Downtime** early this morning due to a minor sensor calibration. While this **Partial System Downtime** was brief, it did affect initial projections for the morning hours. |

*Figure 10.* Example of a generated XLSX layout.

# F. Prompts

## F.1. Semantic Mapping Prompt

**System prompt for Dataset Theming and Semantic Mapping**

You are an expert dataset theming and textualization designer.
Given variables with integer categories, produce one coherent theme.
Return **STRICT JSON** only (no backticks, no commentary).

**UDD Data and Theme**

Output Labels: (categorical) [0, 1, 2]

Variables with category indices:
V0: (categorical) [0, 1, 2, 3]
V1: (categorical) [0, 1, 2, 3, 4]
V2: (categorical) [0, 1, 2]
V3: (numerical) range: [0, 1]

Target Theme: **Philanthropic report**

**JSON Schema Requirements:**
{
"instance_type": "short title",
"theme_summary": "1-3 sentences describing the mapping idea",
"output_labels": { "0": "label name", "1": "label name" },
"output_properties": { "prop": { "0": "label" }, "num": { "type": "numerical", "value_range": {...} } },
"variables": { "<VAR_NAME>": { "role": "...", "categories": { "0": "..." } } },
}

**Constraints:**
1) Cover all listed category indices for each variable
2) Provide short, distinct names for every output label and category
3) Numerical values must include a realistic value_range (min < max)
4) No narrative expansions, no conditionals, and no optional fragments

## F.2. Textual Transformation Prompts

**System prompt for generation of Text samples**

You convert structured assignments into a **1-2** sentence free flowing text.

Rules:
1) Each variable must be included at least once
2) When a variable is included it must be included using the EXACT same phrase as provided
3) You should be creative in the text, but do not add extra facts that can be interpreted as important
4) Output ONLY the final text (no quotes, no JSON, no commentary)

**System prompt for generation of Text Long samples**

You convert structured assignments into a **5-8** sentence free flowing text.

Rules:
1) Each variable must be included at least once, but can occur multiple times
2) When a variable is included it must be included using the EXACT same phrase as provided
3) You should be creative in the text, feel free to add extra context, but not facts that can be interpreted as important
4) Output ONLY the final text (no quotes, no JSON, no commentary)

