# OpenReview forum: "SEDRAS: Symbolically Evaluated Deep Research And Science"
_ICML.cc/2026/Conference — ICML 2026 regular_

### Official Review · Reviewer_6YCX · 2026-03-11

**Soundness:** 4
**Presentation:** 3
**Significance:** 3
**Originality:** 3
**Overall Recommendation:** 5
**Confidence:** 4

**Summary:**

This paper proposes an environment generator for the task of identifying the true data generating function underlying a particular data distribution. This is framed as a task of scientific investigation / experimentation. There is a procedural generator that starts off with a purely mathematical description (input variables, output variables, rules connecting the two) and and then adds semantic flavoring to it based on a theme (e.g. medicine). There are options available for rendering the resulting data distribution into particular formats (e.g. PDF or CSV). The environment that a large language model, "scientist," is placed into can be static, in which case the LLM has to figure out the underlying data distribution using the available data. It can be a more active experimentation setup where the LLM can continually query the environment for new data points, which counts as a form of experimentation. The evaluation is based on having the LLM construct an executable model of the environment, which can then be compared against the true generator of the data distribution to measure accuracy.

**Compliance With Llm Reviewing Policy:**

Affirmed.

**Final Justification:**

The rebuttal reinforced my initial very positive appraisal of the paper. As I wrote in my initial review, I think this is a pretty cool paper that provides a significant and useful resource for AI4Science. I don't see any methodological issues. I carefully read the other reviews (including one Weak Reject), and did not agree with the criticisms raised by the other reviewers. The authors have taken a reasonably principled approach to a hard problem (creating environments to test the ability of LLM agents to do scientific tasks). I'm familiar with the related work in this area and this paper fills a legitimate gap. It's difficult to design a framework that is procedural (so it can generate novel tasks) and complex enough (e.g. going beyond very simplistic templates). I think this paper does a pretty good job of a middle ground and seems like a nice, approachable testbed for people working on agents for scientific tasks (construed broadly).

**Key Questions For Authors:**

I do not have any significant questions for the authors. I think this is a good paper and can be accepted as is. I would prefer for there to be more qualitative examples and for the qualitative examples to be presented more clearly and with an eye towards giving a reader an intuition for what it takes to solve the benchmark. But even if this is not done, I think this paper should be accepted.

**Limitations:**

Yes

**Strengths And Weaknesses:**

I think this is a pretty cool paper that provides a significant and useful resource for AI4Science.

**Soundness**: the environment generation pipeline is principled, and the design makes a lot of sense to me. I like that the authors begin by using a real mathematical structure and then decorate it with semantic content rather than the other way around. This way, it is guaranteed by construction that there is a form of the data generating distribution that can be programmatically checked.

**Presentation**: the writing quality of the paper is good, but I did have quite a bit of difficulty understanding the essence of the environment generation pipeline, what the actually generated scenarios look like, and some sort of high-level overview of what kind of trajectories exist. I do see that there is information provided in the appendix that gives qualitative examples of some of these, but it's not presented in a way that makes it easy for a reader (or a user of the provided environment) to get a sense of what would be required to solve the task.

**Significance**: I think the proposed benchmarks/environments will definitely be useful to people who are working on scientific agents. I am a bit concerned about the difficulty and the rationality of some of these tasks, but I think these concerns might exist in any such benchmark. In particular, the rationality of the task worries me because it's certainly possible, from what I understand about the paper, for the procedural generation to construct a scenario which doesn't make sense in language or strongly violates a realistic prior of a language model while being mathematically valid. It sort of feels like there are going to be some tasks which might be kind of impossible to solve, while there are other tasks which are trivially solved. It would be good to get a better sense of what the distribution of solvability looks like. For example, are there tasks which are not solved by any model?

**Originality**: I don't have many concerns about originality here. I think there could be a strong overlap between these sorts of environments and previous environments for scientific understanding, but I think the procedural generation pipeline is something I haven't seen before for this kind of task. I think there is a nice overlap with computer use agents, where the community is often interested in how agents might work with data that is specified in a variety of formats. Part of the difficulty of the analysis comes from working with data that has been presented in ways that are not trivially usable.

I also think there are some missing references here w.r.t. to _approaches_ for building executable theories of the environment; I don't think these should be _benchmarked_ here but they should be discussed:
- Tang, H., Key, D., & Ellis, K. (2024). Worldcoder, a model-based llm agent: Building world models by writing code and interacting with the environment. In Advances in Neural Information Processing Systems.
- Khan, Z. et al. (2026). One Life to Learn: Inferring Symbolic World Models for Stochastic Environments from Unguided Exploration. In International Conference on Learning Representations.

These are just two papers. You can probably find many others, but the general idea here is to build an executable model of the environment which can then be used for acting. I think the motivation is quite similar in ways to the motivation for this paper, except these sorts of papers are focused on more classic reinforcement learning environments and are more concerned with actually proposing approaches rather than creating test beds.

I also think there are some missing references here w.r.t to procedural generation of environments:
- Executable Functional Abstractions: Inferring Generative Programs for Advanced Math Problems
- GSM-Symbolic: Understanding the Limitations of Mathematical Reasoning in Large Language Models

I think both of these papers, again, have a high degree of conceptual similarity and should probably be discussed. They both start with some sort of underlying mathematical structure and then use LLMs to add semantics to that structure in order to create evaluation benchmarks; they also support procedural generation.

---

> ### Author Rebuttal · Authors · 2026-03-30
>
> Thank you very much for your review and insights. It seems that you share our optimism of this contribution to the AI4Science community.
>
> **Regarding Presentation:**
>
> We do share your thoughts regarding how qualitative examples would improve the readability of the paper. Indeed, there was quite an internal discussion if/how we can squeeze that in given the page limit. In the end we resorted to the appendix, with the motivation that the framework itself has a higher priority, rather than the initial implementation of the representation module.
>
> Given the brief rebuttal period of ICML, it was unfortunately not be possible to make any large sweeping changes to overall formatting of the paper right now. But given that we are accepted, incorporating qualitative examples in the main paper might be possible for the camera-ready version.
>
> **Regarding Significance:**
>
> You correctly observe that the current implementation of the modules may indeed result in “unrealistic” scenarios, where the mathematically valid data violates realistic priors. We would argue that the main contribution of this initial paper is the general framework itself, and that realism is left to future work. Hopefully, the modularized nature will allow people to implement more realistic scenarios, and ideally share those with the broader AI4Science community.
>
> For example, one can easily imagine a workshop-level paper that provides a strong SEDRAS pipeline for Legal Cases, Medical Diagnostics, etc… What we provide is a modularized framework, along with the novel idea of evaluation via the theory-conversion loop.
>
> Furthermore, we would argue that “unrealistic” scenarios are themself appealing, given that we are aware of this. The more counter-intuitive the scenarios are compared to the real-world, the harder it might become to solve for certain models. Simply put, this would allow one to measure how biased a system is towards memorized prior information about the world, and to what extent it is capable of discard those priors and reason from only the provided data.
>
> **Regarding Originality / Related Work:**
>
> You raise some valid related work papers, and we have tried to incorporate them as best we can in the main paper, given the space limitations. In particular, thehe GSM-Symbolic paper in the introduction section. Additionally, we have created an “extended related works section” in the appendix, that  covers the majority of the references you mention. But we refrained from citing “Executable Functional Abstractions”, as it seems it was submitted to ICLR this year (and rejected).
>
> Furthermore, reviewer G5HB provided several good papers that we have factored in. In particular, the HypoBench paper is very relevant as it provides a rudimental bottom-up framework for synthetic data. Therefore we have restructured our framing and nattarive slightly. Please see our response to reviewer G5HB for more information on this.

---

> > ### Author Rebuttal · Reviewer_6YCX · 2026-03-31
> >
> > My initial review was very positive, and I saw no major issues with the paper.

---

### Official Review · Reviewer_LciP · 2026-03-12

**Soundness:** 2
**Presentation:** 3
**Significance:** 2
**Originality:** 3
**Overall Recommendation:** 3
**Confidence:** 4

**Summary:**

This paper introduces a novel evaluation framework called SEDRAS, designed to assess the capabilities of large language models (LLMs) in In-context Scientific Discovery and deep research tasks. It addresses issues in existing benchmarks, such as susceptibility to data contamination, lack of interactivity, and difficulty in measuring the model’s true inductive generalization ability.

**Compliance With Llm Reviewing Policy:**

Affirmed.

**Final Justification:**

Most of my concerns have been addressed, but I still remain skeptical about the evaluation gap between generated problems and real-world problems, so I am maintaining my negative assessment.

**Key Questions For Authors:**

1. Why assess the capabilities of closed-source models? How do open-source models perform on the current benchmark?

2. How can it be ensured that these generated test samples accurately reflect the model’s capabilities in a real scientific research environment?

3. Since this paper uses a generative approach to construct the test benchmark, could different generation batches significantly affect the results? In other words, is the model’s performance comparable across different generated datasets?

Other questions can be found in the Weaknesses section. If they are reasonably addressed, I would consider increasing my score.

**Limitations:**

yes

**Strengths And Weaknesses:**

Strengths:

1. By using procedurally generated synthetic scenarios, it fundamentally avoids the risk of the model having been exposed to the test data during training.

2. SEDRAS-2026 includes 600 scenarios, covering a diverse range of tasks from simple classification to complex rule-based challenges.

3. The paper uses numerous flowcharts to facilitate understanding, allowing readers to quickly grasp the proposed method.

Weaknesses:

1. Although UDD can be quite complex, it is essentially based on rule-based symbolic logic and cannot fully replicate real scientific experiments, nor can it simulate authentic scientific discovery scenarios.

2. It is highly dependent on the model’s coding ability. If a model has strong scientific intuition but relatively weaker coding skills, its theories may be seriously misinterpreted, resulting in lower scores.

3. During testing, the model’s abilities are evaluated using test cases, but the correctness of a limited number of test examples cannot accurately reflect the theoretical correctness of the model.

---

> ### Author Rebuttal · Authors · 2026-03-30
>
> Thank you very much for your feedback and questions. We have updated the paper accordingly and hope to clarify some of your questions here.
>
> **Weakness 1: Rule-based UDD**
>
> As we state in the paper, our UDD generation is simply an arbitrary starting point. The UDD does not need to be rule-based and can be implemented in many possible ways. As we discuss in Section 5, one could model the UDD using sophisticated differential equations or even probabilistic neural networks.
>
> **Weakness 2: Coding abilities and end-2-end evaluation**
>
> Questioning the design choice of end-2-end evaluation is very valid, and we have made updates to Introduction, Section 3 to clarify that this does not need to be the case.
>
> Our intention is not to strictly force users to always have the same model act as the different components of the scenarios. If one wishes to push specifically for the theory generation, one could imagine an alternative “SEDRAS 2026 GemTheory” version of the dataset, where Gemini 3.0 always acts as the theory converter. Indeed, we have now added an Appendix section that shows how the results would differ if that would have been the case.
>
> Furthermore, we argue that a major strength of our framework is the ease with how such modification can be made. We welcome whatever modifications and versions of the framework that future work sees most suitable for their scenarios.
>
> In a sense our decision to have the same model act as all LLM components of SEDRAS 2026 dataset, was to hedge against potential reviewer concerns of the validity of the evaluation. As one might be concerned that the theory converter itself introduces too much noise. Instead formulating the evaluation, as the full evaluation loop itself, mitigates such concerns as any LLM error that arises is a fault of the model being tested, and should therefore be rightfully represented in the evaluation results.
>
> **Weakness 3: finite test cases**
>
> Thank you for your thoughts around the finite nature of the test cases used to evaluate each theory, and we have updated the paper to clarify this further. We use 1000 test cases to evaluate the accuracy of each converted theory. Thanks to the Central Limit Theorem, this number of test-cases combined with the number of scenarios results in a *very* robust evaluation, where even small differences in results can be considered statistically significant.
>
> Indeed, we have created the new Appendix A.2, which visualizes how an accuracy difference of 1% yields a P-value <0.01, when the number of test-cases are 1000, with even as few as 10 scenarios. Thus, we would strongly argue against your point here, as our experiments include far more than 10 scenarios.
>
> Finally, we would also bring to your attention that for the lower data complexity levels (1, 2) the 1000 randomly selected test-cases encompass every possible data point for that scenario. As such, for UDDs these the evaluation is a perfect and exhaustive test of the theory.
>
> **Question 1: Closed source models**
>
> Here, we would ask you for clarification: Are you asking why we **only** assess the capabilities of closed-sourced models? If so, the main contribution of the paper is the evaluation framework itself as opposed to the evaluation results on currently available models. Given this, and the ever growing list of available models, we decided to draw the line at popular closed-source models.
>
> The open-benchmark platform that we provide also allows anyone to benchmark their own model. In practice, this is achieved by the user setting up an accessible inference end-point, and then our open SEDRAS framework will iteratively query it and evaluate the model. This should then automatically update the leaderboard.
>
>
> **Question 2: Transferability to scientific domains**
>
> Domain specific expertise will be required to produce a specialised SEDRAS pipeline that matches a given scientific domain. Simply put, what we provide is the general “simulation framework” for efficient LLM evaluation. Specific simulation dynamics would need to be implemented for specific scenarios.
>
> For example, one can easily imagine a workshop-level paper that provides a strong SEDRAS pipeline for Legal Cases, Medical Diagnostics, etc… What we provide is a modularized framework, along with the novel idea of evaluation via the theory-conversion loop.
>
> Furthermore, we also find the idea of creating “non-scientifically” accurate scenarios very appealing. The more counter-intuitive the scenarios are compared to the real-world, the harder it might become for certain models. Simply put, this would allow one to measure how biased a system is towards memorized prior information about the world, and to what extent it is capable of discard those priors and reason from only the provided data.
>
> **Question 3: Generation Batch Stability**
>
> From our initial experiments we found that the average accuracy between different generation batches remains fairly stable.

---

> > ### Author Rebuttal · Reviewer_LciP · 2026-04-03
> >
> > The authors argue that their theoretical framework can be quickly transferred to other domains, but they provide no real case studies to support this claim. For example, they do not demonstrate that, after adapting the framework to a specific scientific domain, it can indeed be used to evaluate a model’s actual capability in that domain.
> >
> > For a paper centered on evaluation, this lack of concrete evidence is not sufficient.
> >
> > Therefore, I am inclined to keep a negative score.

---

> > > ### Author Response · Authors · 2026-04-05
> > >
> > > You are correct in that we currently don’t provide an implementation that directly demonstrates downstream skill transferability. However, we would argue that this is beside the main point of our paper, and zooming in on this single property misses the larger picture. Our core contribution is a principled framework to evaluate the “general” scientific capabilities of a model, and the framework's extendability is more a hint towards its potential longevity.
> > >
> > > **Abstract Datasets**
> > >
> > > Principled datasets that have no direct link to downstream tasks are extremely valuable. If designed properly, they can help diagnose fundamental model properties, *exactly because* they are not tied to any specific downstream task. Indeed, a model can exhibit strong performance in a specific domain, yet fail in the more general sense. **For the community interested in achieving general AI-Scientists**, SEDRAS is therefore a major contribution.
> > >
> > > Consider for example datasets like ARC for evaluating fundamental reasoning in an “unpolluted” way. Finding that, for example GPT-4, scores poorly on ARC does not entail that GPT-4 is useless when applied to a real scenario. But it helps demonstrate that methods and models are less general than one might have originally thought. And achieving generality and robustness is very much a worthwhile goal in the field of Artificial Intelligence, but not necessarily the same goal as immediately looking to maximize a specific downstream task.
> > >
> > > What we provide is a principled bottom-up approach to generating “general” rule-discovery scenarios. So even discarding implementation support of "realistic" settings, it inherently allows us to analyze models' general rule-discovery capabilities, without the bias of any given domain. As demonstrated in the paper, this analysis can be very fine-grained and enables one to pinpoint strengths and weaknesses of different models.
> > >
> > > **Findings in SEDRAS-2026**
> > >
> > > In SEDRAS 2026, we can clearly see how the capabilities change depending on how we move the different decoupled axes. And this can already be used to demonstrate shortcomings in our current models. Our paper highlights many of these findings, but consider for example the following:
> > >
> > > - Altering the representations of the data should not result in a decreased capabilities. However, all models perform significantly worse when the data is represented via files.
> > > - We can see how Gemini actually performs significantly worse on the RAW representations, compared to its own performance on the textual representations. With GPT having the exact opposite problem, to an even larger extent.
> > > - Section 4.3 demonstrates that none of the tested models perform well on active exploration, and performs worse than random exploration. Intuitively, a strong general scientist should be able to determine what new data which might be useful, and certainly not be worse than random.
> > >
> > > These are already valuable findings. The fact that they are not tied to a given specific domain are not detrimental to their impact.
> > >
> > >
> > > **Conclusion & Paper Changes**
> > >
> > > We thank you for prompting us to clarify these contributions further.
> > > The introduction, discussion and conclusion, have now been updated to further clarify the appeal of a fully abstract evaluation framework. Making it clear that although SEDRAS is yet to be implemented on any specific domain, it is already providing clear indications of weaknesses that need to be covered for a truly general AI-Scientist (or AGI if you will). With SEDRAS-2026 demonstrating how model behavior degrades as certain aspects of the scenarios change, contrary to how an ideal model would behave.

---

### Official Review · Reviewer_G5HB · 2026-03-12

**Soundness:** 3
**Presentation:** 3
**Significance:** 3
**Originality:** 2
**Overall Recommendation:** 4
**Confidence:** 3

**Summary:**

This paper introduces SEDRAS, a modular benchmark framework for evaluating LLMs on scientific-discovery-style reasoning under controllable synthetic settings. The framework factorizes task difficulty along three axes: the complexity of the underlying data distribution (UDD), the complexity of the representation exposed to the model, and the complexity of the interaction dynamics through which the model can inspect or explore the environment. In each scenario, the model is asked to infer an interpretable theory in natural language and then convert that theory into executable code, which is evaluated programmatically against the hidden ground-truth distribution. The authors instantiate this framework as SEDRAS-2026, a suite of 600 procedurally generated scenarios, and evaluate several frontier LLMs. A key empirical finding is that current models perform much better in static settings than in active exploration settings, suggesting that strategic information gathering remains a major weakness.

**Compliance With Llm Reviewing Policy:**

Affirmed.

**Final Justification:**

Thank you for the rebuttal, I keep my original score.

**Key Questions For Authors:**

1. My main reservation concerns positioning and originality. The paper would benefit from a sharper comparison to the closest recent benchmark literature, especially DiscoveryBench, Auto-Bench, ResearchBench, HypoBench, and LLM-SRBench. Relative to these works, what do the authors see as the primary novel contribution of SEDRAS? A convincing clarification here could improve my originality assessment.

2. The current framing sometimes suggests a broad benchmark for “deep research,” while the actual setup seems especially strong as a controllable benchmark for hidden-rule discovery under representation and interaction shifts. Can the authors clarify the intended scope of the benchmark, and explain why this is the right abstraction for the capability they want to measure? A tighter framing could improve my assessment of significance and presentation.

3. More specifically, among the paper’s ingredients, which one do the authors regard as the main conceptual contribution: (i) the three-axis factorization of benchmark difficulty, (ii) the theory-to-code validation loop, or (iii) the end-to-end evaluation of discovery under controllable synthetic generation? Clarifying this would help me better judge the paper’s novelty.

4. The benchmark evaluates the subject model end-to-end across theory generation, theory conversion, and in some settings experiment-query translation. Could the authors more explicitly justify this design choice as part of the benchmark target, rather than an incidental conflation of sub-tasks? A stronger justification would increase my confidence in the benchmark’s validity.

**Limitations:**

yes

**Strengths And Weaknesses:**

Strengths:

- The paper targets an important and timely capability. As LLM systems are increasingly framed as scientific assistants or deep-research agents, evaluation settings that go beyond static contaminated benchmarks are valuable.
- The benchmark design is thoughtful. In particular, separating difficulty into UDD complexity, representation complexity, and interaction dynamics provides a useful diagnostic decomposition that is often missing in existing benchmarks.
- The theory-to-code evaluation loop is a meaningful contribution. It tests not only whether the model can produce plausible natural-language explanations, but whether those explanations are precise enough to be operationalized and checked against hidden ground truth.
- The experimental analysis is informative. Beyond overall ranking, the paper identifies a clear and practically relevant failure mode: current frontier models remain substantially weaker in active, self-directed exploration than in static settings.

Weaknesses:

- The paper’s originality is moderate rather than high, and the current positioning somewhat overstates the novelty. In my view, the main contribution lies in the benchmark design and in the decomposition of difficulty, rather than in a fundamentally new evaluation paradigm.
- The related-work discussion should be sharper with respect to the closest neighboring benchmarks. In particular, the paper would benefit from a more explicit comparison to recent work such as DiscoveryBench (towards data-driven discovery with LLMs), Auto-Bench (automated benchmark for scientific discovery in LLMs), ResearchBench and HypoBench (hypothesis/discovery benchmarking), and LLM-SRBench (scientific equation discovery). These works differ in scope, but they are directly relevant to the paper’s novelty claims.
- Because of these missing comparisons, the current framing around being a uniquely positioned benchmark at the intersection of scientific discovery, interaction, and deep research is not yet fully convincing. The paper should better clarify which aspects are genuinely new: e.g., the specific three-axis factorization, the theory-to-code validation loop, or the particular controllable synthetic setting.
- The “deep research” framing is somewhat broader than the actual evaluation setup. In practice, SEDRAS seems best understood as a controllable benchmark for hidden-rule discovery under representation and interaction shifts, rather than a full proxy for realistic open-web, citation-grounded deep research. This does not remove the value of the benchmark, but it does affect how its significance and novelty should be interpreted.
- The benchmark intentionally evaluates the model end-to-end across multiple roles (theory generation, theory conversion, and in some settings experiment-query translation). This is a defensible design choice, but it also entangles several sub-capabilities, and the tradeoff should be discussed more explicitly.

Overall by dimension:

- Soundness: good for a benchmark paper; the design is coherent and the main empirical claims are reasonably supported.
- Presentation: good overall, though the related-work positioning and scope claims should be tightened.
- Significance: good, because the target capability matters and the benchmark offers useful diagnostic value.
- Originality: fair; the most convincing novelty lies in the benchmark decomposition and validation pipeline, not in an entirely new benchmark category.

---

> ### Author Rebuttal · Authors · 2026-03-30
>
> Thank you for your thoughtful review, and the related work papers you bring to our attention.
> We have updated the introduction, related work and discussion sections accordingly.
> Furthermore, we have created an “extended related works section” in the appendix, that covers all of the references you mention.
>
>
> **Question 3: Primary contributions**
>
> As you have correctly pointed out, our main novel contribution is the bottom-up and modularized design philosophy. In particular this enables the following unique properties:
>
>
> Fine-grained control, and analysis, of task scalability along our three independent dimensions. (UDD, Representation, Interaction)
> Generating scenarios that combine both rich data representations and interactive scenarios. This is something that none of the related work benchmarks are able to do (as to our knowledge at least).
>
> Furthermore, due to the modularization, we argue that SEDRAS can scale further and to more varied scenarios than existing benchmarks. And equally important, should make it easier for future work to extend and build upon.
>
>
> **Question 1 & 2: Framing and positioning**
>
> We understand your main reservation around positioning, and given the HypoBench paper you present we have toned-down claims of being “first bottom-up” benchmark. Following are our thoughts on the particular benchmarks you mentioned, two of which we have managed to squeeze into the main paper’s related work section. However all the mentioned papers are now present in the appendix “extended related works section”.
>
>
> **HypoBench: (added to related work)**
> This paper does indeed provide a bottom-up framework that generates textual samples from a rule-based UDD. We are grateful that you have brought it to our attention. There are however two particular ways in which SEDRAS improves upon this idea. (in addition to the points mentioned above)
>
> 1 - HypoBench only provides static dataset tasks. And it is not immediately clear how one would extend it to support more dynamic scenarios.
>
> 2 - The representation pipeline for HypoBench is comparably rudimental. Samples are represented by inserting its corresponding keyword into an LLM generated textual template. This arguably results in low variance between samples, but most importantly provides no initial controllability for the representation complexity itself.
>
> Even our initial representation pipeline pushes way beyond this. For each sample we generate a unique natural language text, and provide control for both the length and how samples are rendered/formatted. Resulting in a controllable complexity ranging from RAW variable assignments to dynamically formatted PDF/Excel files.
>
> **AutoBench:  (added to related work)**
>
> AutoBench focuses on graph-based discovery tasks with a fixed interaction loop, where data is generated through model-driven queries. While this enables structured exploration, it operates over a narrow task class and does not decouple the underlying data, representation, and interaction dynamics.
>
> **DiscoveryBench:**
>
> DiscoveryBench evaluates hypothesis formation over real and semi-synthetic datasets, where tasks follow a fixed pipeline with multiple valid outputs. Our framework, whilst allowing for multiple valid outputs, targets instead recovery of the underlying generative mechanism under a fully procedural and agentic setting.
>
> **ResearchBench**
>
> ResearchBench constructs scientific discovery tasks from real-world literature, prioritizing novelty beyond pretraining data. However, its reliance on fixed corpora limits controllability and comparability across models, in contrast to our fully procedural and endlessly extensible setup.
>
>
> **LLM-SRBench**
>
> LLM-SRBench focuses on symbolic regression over numerical data, relying on external evaluators for correctness. In contrast, our framework supports broader modalities and enables direct programmatic evaluation against known ground truth.
>
> **Question 4: End-to-end Evaluation**
>
> Questioning the design choice of end-2-end evaluation is very valid, and we have made updates to Introduction, Section 3 to clarify that this does not need to be the case.
>
> Our intention is not to strictly force users to always have the same model act as the different components of the scenarios. If one wishes to push specifically for the theory generation, one could imagine an alternative “SEDRAS 2026 GemTheory” version of the dataset, where Gemini 3.0 always acts as the theory converter. Indeed, we have now added an Appendix section that shows how the results would differ if that would have been the case.
>
> Furthermore, we argue that a major strength of our framework is the ease with how such modification can be made. We welcome whatever modifications and versions of the framework that future work sees most suitable for their scenarios.
>
> In a sense our decision to have the same model act as all LLM components of SEDRAS 2026, was to hedge against potential concerns of the evaluation validity, as discussed in Section 5.1.

---

> > ### Author Rebuttal · Reviewer_G5HB · 2026-04-03
> >
> > Thank you for the detailed rebuttal. The rebuttal strengthens the paper. My main residual reservation is still that the novelty is moderate rather than high, and that the “deep research” framing remains somewhat broader than what is directly evaluated. However, I believe the paper makes a useful benchmark contribution, and I keep my original score.

---

> > > ### Author Response · Authors · 2026-04-08
> > >
> > > Again, we thank you for highlighting the related work which has improved our framing.
> > > Seeing that you find our contribution to be a useful benchmark contribution, allow us to make a final attempt to convince you to increase your recommendation.
> > >
> > > **DeepResearch**
> > >
> > > You raise a fair point regarding the term “Deep Research”, and in all honesty it is such a modern “buzzy” term that it may, or may not be correct for our framing. From our understanding the term originated from OpenAI when they released it as a mode for ChatGPT, and is not very clearly defined. So it could be that another term would be more suitable.
> > >
> > > However, our framework does provide scenarios where the model needs to interact with files, parse data, run experiments etc. Something we would argue falls into the category Deep Research. Even more so, this work provides a foundation for implementing such scenarios, with all the benefits of our framework, such as the fine-grained analysis and memorization resistance
> > >
> > > All that being said. If you have an alternative term that covers such scenarios, we would gladly concede and rename/reframe the paper accordingly.
> > >
> > > **Contribution and Novelty**
> > >
> > > Besides the general quality of our framework, and how it goes significantly beyond what any prior bottom-up framework has done, we would like to highlight the specific facets of our contribution that we believe elevate the novelty and utility of SEDRAS within the current research landscape:
> > >
> > >
> > > - Multidimensional Scalability: Unlike existing benchmarks that scale difficulty along a single axis, SEDRAS allows for independent and fine-grained control over the logic of the data (UDD), the complexity of its presentation (Representation), and the depth of the interaction protocol (Dynamics).
> > > - Fine-Grained Diagnostic Analysis: Our modularized design enables researchers to perform a "post-mortem" on model failures to determine exactly where reasoning breaks down, whether it is a struggle with high-dimensional logic, obfuscated file formats like PDFs and XLSX, or strategic information gathering.
> > > - Unique Scenario Convergence: SEDRAS is the first scientific-oriented framework to simultaneously combine rich, multi-modal data representations with interactive, agentic scenarios.
> > > We believe these contributions represent a significant step toward a more robust and longitudinal understanding of AI progress in scientific discovery.
> > >
> > > **Conclusion**
> > >
> > > Given that the points above are truly novel, and that we provide a strong step further in the directions where existing works already exist, would you consider increasing your recommendation? In any case, we thank you for engaging with our work and your encouragements.

---

### Decision · Program_Chairs · 2026-04-30

**Decision:**

Accept (regular)

**Comment:**

SEDRAS introduces a modular evaluation framework for assessing LLMs on in-context scientific discovery tasks, with controllable complexity along three independent axes: underlying data distribution logic, representation format, and interaction dynamics. The framework uses a theory-to-code validation loop where models must produce executable theories that are programmatically compared against ground-truth distributions. An initial suite of 600 scenarios is used to evaluate frontier LLMs, revealing that all models struggle significantly with active exploration compared to static settings.
Reviewer 6YCX recommended acceptance, praising the principled environment generation pipeline, the procedural generation approach as filling a legitimate gap in the AI4Science evaluation landscape, and the conceptual contribution of starting from mathematical structure and decorating with semantics. Reviewer G5HB gave a weak accept, finding the three-axis decomposition and theory-to-code loop valuable while noting that the novelty is moderate and the "deep research" framing is broader than what is directly evaluated. Reviewer LciP gave a weak reject, primarily concerned about the lack of demonstrated transferability to real scientific domains. The authors clarified that the core contribution is a domain-agnostic evaluation framework, analogous to ARC for reasoning, and that its value lies precisely in diagnosing fundamental model capabilities without domain-specific confounds. I have read the authors' rebuttals and AC-confidential comments and find that the initial concerns raised by Reviewer LciP were largely clarified through the rebuttal process, though the reviewer maintained their score.
The paper makes a useful contribution as an evaluation framework with a principled design, and the finding that current models perform worse than random baselines on active exploration is noteworthy. The authors should tighten the "deep research" framing, add more qualitative examples to the main text, and incorporate the related work comparisons from the rebuttal into the camera-ready version.